# Graphene Oxide Nanoparticles and Organoids: A Prospective Advanced Model for Pancreatic Cancer Research

**DOI:** 10.3390/ijms25021066

**Published:** 2024-01-15

**Authors:** Shaoshan Mai, Iwona Inkielewicz-Stepniak

**Affiliations:** Department of Pharmaceutical Pathophysiology, Faculty of Pharmacy, Medical University of Gdańsk, 80-210 Gdańsk, Poland; shaoshan.mai@gumed.edu.pl

**Keywords:** graphene, graphene oxide, 3D culture, organoids, pancreatic cancer

## Abstract

Pancreatic cancer, notorious for its grim 10% five-year survival rate, poses significant clinical challenges, largely due to late-stage diagnosis and limited therapeutic options. This review delves into the generation of organoids, including those derived from resected tissues, biopsies, pluripotent stem cells, and adult stem cells, as well as the advancements in 3D printing. It explores the complexities of the tumor microenvironment, emphasizing culture media, the integration of non-neoplastic cells, and angiogenesis. Additionally, the review examines the multifaceted properties of graphene oxide (GO), such as its mechanical, thermal, electrical, chemical, and optical attributes, and their implications in cancer diagnostics and therapeutics. GO’s unique properties facilitate its interaction with tumors, allowing targeted drug delivery and enhanced imaging for early detection and treatment. The integration of GO with 3D cultured organoid systems, particularly in pancreatic cancer research, is critically analyzed, highlighting current limitations and future potential. This innovative approach has the promise to transform personalized medicine, improve drug screening efficiency, and aid biomarker discovery in this aggressive disease. Through this review, we offer a balanced perspective on the advancements and future prospects in pancreatic cancer research, harnessing the potential of organoids and GO.

## 1. Introduction

Pancreatic ductal adenocarcinoma (PDAC) is one of the most aggressive and deadliest forms of cancer, known for its rapid progression and poor prognosis. Despite significant advancements in cancer research and treatment modalities, PDAC continues to have a dismal five-year survival rate of less than 11%, with this figure dropping to less than 3% in patients with advanced stages of the disease [1]. This high mortality rate is attributed to several factors including the cancer’s high malignancy, insidious onset, lack of distinct symptoms, challenging anatomical location, low resection rate, and high recurrence rate.

Common signs and symptoms of PDAC, often appearing only in advanced stages, include jaundice, abdominal and back pain, unexplained weight loss, and digestive difficulties. Globally, pancreatic cancer is the seventh leading cause of cancer-related deaths, with its incidence and mortality rates closely aligning due to its aggressive nature. According to the World Health Organization, there were over 450,000 new cases and 430,000 deaths worldwide in 2020, reflecting its substantial impact on global health [2,3].

Currently, standard treatment modalities for PDAC include surgery, chemotherapy, and radiation therapy. While surgical resection offers the best chance for a cure, only a small fraction of patients is eligible for this option. The majority of pancreatic cancer cases are diagnosed at an advanced stage, rendering surgery infeasible [4]. The mainstay of treatment for advanced pancreatic cancer, chemotherapy, has limited efficacy. Drugs such as gemcitabine have been the standard, but they offer only modest improvements in survival and are frequently associated with significant side effects. In addition, as it is a broadly used drug, almost all pancreatic cancer patients eventually develop resistance to this drug [5]. FOLFIRINOX (oxaliplatin, irinotecan, leucovorin, 5-fluorouracil) could directly improve the overall survival (OS) rate of patients with metastatic pancreatic tumor (HR 0.76, 95% Cl 0.67–0.86, *p* < 0.001) but had no benefit on progression-free survival (PFS) [6]. Standard radiotherapy options, which typically deliver 40 to 60 Gy in 1.8–2.0 Gy fractions, offer minimal to no survival benefit for patients with locally advanced unresectable pancreatic cancer (LAPC) who have undergone chemotherapy [7]. Its role is limited due to the proximity of the pancreas to critical organs, which increases the risk of damage to surrounding tissues. Therefore, these limitations of treatment underscore the urgent need for innovative therapeutic strategies.

In the realm of research, organoid models have emerged as a promising tool. These organoids accurately represent the genomic, proteomic, and morphological characteristics of parental tumors, providing a more faithful replication of patient tumors compared to traditional in vitro 2D cell cultures [8]. Organoids are advantageous for their ability to be passaged long-term, cryopreserved, and genetically manipulated without significant genomic or epigenetic alterations, making them ideal for constructing living biobanks and advancing personalized therapy approaches. These organoid models have been instrumental in understanding disease progression, studying tumor microenvironment interactions, and contributing to precision medicine [9,10].

Graphene oxide (GO), a derivative of graphene, stands out in the realm of nanomaterials due to its unique properties and potential applications, particularly in the biomedical field. GO is characterized by its chemical and mechanical stability, biocompatibility, and a two-dimensional structure that allows for extensive surface modification [11]. These modifications can be tailored with various functional groups such as epoxide, hydroxyl, and carboxyl, enabling the attachment of biomolecules such as proteins, DNA, and RNA [12]. This adaptability makes GO an attractive candidate for a range of applications, including drug and gene delivery, phototherapy, and bioimaging. In medical research, GO’s ability to be used in drug delivery systems, diagnostics, tissue engineering, and gene transfection is particularly noteworthy [13]. Its solubility in water and intrinsic fluorescence properties in the visible/near-infrared spectrum enhance its suitability for these applications [14,15]. Recent studies have also explored the possibility of using GO for specific targeting of cancer cells, potentially opening new avenues for cancer therapy, including the inhibition of metastasis.

In the context of tumor therapy, GO is often used as a nano-carrier due to its ability to load a large number of hydrophobic drugs containing benzene rings. Its nano-network structure and hydrophobicity play a crucial role in this process [16,17]. Additionally, GO’s surface properties can be affected by pH changes [18]. It remains stable at a neutral pH but becomes less stable at more extreme pH values [19]. This characteristic is particularly useful in cancer therapy, as tumor tissues generally have a more acidic microenvironment compared to normal tissues. Therefore, at lower pH values, such as those found in tumor tissues, the protonation weakens the hydrogen bond interaction between the drug and GO. This pH-sensitive property can be incorporated into the design of an anticancer drug delivery system, making GO an intelligent nano-carrier that releases drugs more effectively in the acidic environment of tumor tissues [20]. In addition, it is often functionalized with various components, such as synthetic polymers such as polyethylene glycol (PEG) or natural polymers [21]. This functionalization can be achieved through covalent modification, which may alter the original structure of GO, or non-covalent methods, which do not affect its native structure. These modifications enhance the stability of GO in physiological solutions and facilitate its use in drug delivery [22].

In this review, we aim to provide a comprehensive overview of the recent developments in organoids (Figure 1) and GO, focusing on their roles in restructuring the cancer microenvironment, their advancements, and potential biomedical applications. We consolidate current knowledge on the properties and applications of organoids and GO (Figure 2) in cancer research. This includes discussing the ongoing challenges in these fields and exploring the potential synergistic future of organoids and GO platforms. Our primary focus was on their application in personalized treatment strategies for pancreatic cancer, emphasizing the need to address current limitations to fully realize their potential in this context.

## 2. Organoid Models for Studying Cancer

### 2.1. Origin of Organoids

#### 2.1.1. Organoids Generation from Resected Tissues

Over the last decade, most origins of cancer organoids have been derived from primary tumors and metastatic lesions. They are collected from surgical resections. Sato et al. described the generation and expansion of patient-derived organoids from colon tissue [23]. Pancreatic cancer organoids were built to reveal genes and pathways altered during disease progression [24]. Lung cancer organoids recapitulated the tissue architecture of the primary lung tumors [25]. To date, these cancer organoids include colorectal [26,27], ovarian [28,29,30], prostate [31,32], pancreatic [33,34], liver [35,36], bladder [37,38], lung [39,40,41], gastric [42,43], brain [44,45], esophagus [46,47], and endometrium [48,49,50,51].

#### 2.1.2. Organoids Generation from Biopsy

The one of main challenges of organoids is tissue accessibility. Except for small fragments of surgically resected tissue, solid and liquid biopsies are used as sources of tumor organoid cultures. For example, bladder cancer organoid lines can be established efficiently from patient biopsies acquired before and after disease recurrence [52]. Human prostate cancer organoids were retrieved from biopsy specimens and circulating tumor cells successfully in long-term cultures [53]. Protocols for establishing organoids from human ovarian cancer biopsies have been developed [54,55]. In addition, colorectal organoids can be derived from metastatic biopsy specimens with a high success rate and genetically represent the metastasis they were derived from [56]. Circulating tumor cells (CTCs) are a rare subset of cells found in the blood of patients with solid tumors; the CTC-derived pre-clinical model matches the mutation with the primary tumor, which could be applied in clinical research for the evaluation of disease progress [57].

#### 2.1.3. Organoid Generation from Pluripotent Stem Cells

Organoids developed from human pluripotent stem cells (PSCs) are particularly useful for tissue shortage, such as the nervous system and retina [58,59]. Human-PSC-derived organoids with components of all three germ layers (ectoderm, mesoderm, endoderm) have been generated, resulting in the establishment of a new human model system. hPSCs, including human induced pluripotent stem cells (hiPSCs) and human embryonic stem cells (hESCs). Both of them share an unlimited proliferative ability and the developmental potential to generate all three germ layers [60]. Colonic organoids derived from hiPSC for modeling colorectal cancer provide an efficient strategy for drug testing [61]. Patient-derived iPSCs from a Li-Fraumeni syndrome (LFS) family were used to investigate the role of mutant p53 in the development of osteosarcoma (OS) [62]. iPSC-derived embryoid bodies of a patient with hereditary c-met-mutated papillary renal cell carcinoma (PRCC) were generated spontaneous aggregates organizing in structures that expressed kidney markers such as PODXL and Six2 [63]. iPSCs were differentiated into pancreatic ductal and acinar organoids that recapitulate the properties of the neonatal exocrine pancreas, as well as the mutated KRAS G21D oncogene in organoids to form pancreatic cancer [64]. Engineering prostate cancer from iPSC is used to develop preclinical tools in prostate cancer studies [65].

#### 2.1.4. Organoids Generation from Organ-Specific Adult Stem Cells

Adult stem cell (ASC)-derived organoids were shown to be genetically stable over long periods [66]. The genetic modification of small intestine, stomach, liver, and pancreas organoids has opened up avenues for the manipulation of ASC in vitro, which could facilitate the study of human biology and allow gene correction for regenerative medicine purposes [67]. Human intestinal stem cells mutated via CRISPR/Cas9 technology for colorectal cancer organoid cultures remain genetically and phenotypically stable [68]. Efficient protocols for the culture of breast cancer (BC) organoids that are established from mammary epithelium serve as a representative collection of well-characterized BC organoids for BC research and drug discovery [69]. On exposure to pregnancy signals, endometrial organoids derived from ASC develop the characteristics of early pregnancy [70].

#### 2.1.5. Organoid Generation from 3D Printing

The lack of precise architectures and size are the key limitations of tissuee- and PSC-derived organoids. 3D bioprinting of organoids has shown its advantages in recent years. It is a promising technology to precisely position the biological materials, living cells, and growth factors for the computer-aided generation of bioengineered structures. Bioinks and bioprinters are two necessary and key factors used in 3D printing. Natural polymer-based bioinks such as agarose [71], alginate [72], collagen [73], hyaluronic acid [74], gelatin, and matrigel [75] are used for printing. Bioprinting methods include inkjet printing (nozzle-based), extrusion bioprinting (nozzle-based), light-based bioprinting stereolothography, digital light processing (nozzle-free), and laser-assisted bioprinting (nozzle-free).

It is difficult to restructure pathological cancer tissue in organoids and standardize the number and size of organoids in current organoid culturing. However, 3D printing is a prospective tool for cancer organoid treatment. Bioprinting can reconstitute the tumor microenvironment, for example, control matrix properties and integrate vascular networks. Glioblastoma-on-a-chip is a 3D bioprinting model in which patient-derived glioblastoma cells are co-cultured with endothelial cells to form a cancer–stroma concentric ring structure, which serves the purpose of screening for effective treatment modalities for patients [76]. In a 3D bioprinting collagen model, biomimetic chimeric organoids were developed by co-bioprinting breast cancer cells and normal mammary epithelial cells, which showed a more significant increase in 5-hydroxymethylcytosine in the chimeric structure than the tumoroid one [77]. A 3D bioprinting organotypic pancreatic adenocarcinoma model was used to restructure a cancer microenvironment that consisted of endothelial and pancreatic stellate cells [78]. A high-throughput bioprinting system was used to construct a Matrigel substrate via co-bioprinting ovarian cancer cells (OVCAR-5) and fibroblasts (MRC-5) [79]. Hela cells and gelatin/alginate/fibrinogen hydrogels have been used to construct cervical tumor models in vitro via extrusion-based bioprinting [80]. A hepatic cancer model was fabricated through a two-step stereolithographic bioprinting method [81].

### 2.2. Extracellular Matrix of Organoids

#### 2.2.1. Animal-Derived Matrix

The extracellular matrix (ECM) is a dynamic polymer network used to influence the cell biology of neoplastic and tumor microenvironments (TMEs). Hydrogels derived from the decellularized basement membrane of murine Engelbreth-Holm-Swarm (EHS) sarcoma, such as Matrigel or Geltrex, are the most commonly used matrix for organoid culturing [82,83]. EHS matrices have been widely adopted in cancer organoids, relying on their potential to recapitulate the 3D tumor structure and provide growth factors that maintain TME. For human cancer organoid culturing, however, the EHS matrix is limited by its ill-defined, poorly tunable animal-derived scaffolds and batch-to-batch variability [84,85]. It is associated with increased collagen deposition and remodeling in tumor progression. Collagen type I matrices are also a less expensive alternative for cancer organoids. Similarly, animal sources that face the same problem as EHS matrices, and their microstructures (for example, fibril diameter, and alignment) are highly dependent on the rate of pH and temperature changes during gelation [86,87].

#### 2.2.2. Engineered Matrix

Engineered matrices have yet to be routinely applied to human cancer organoid cultures; however, compared with animal-derived matrices, they offer high batch-to- batch reproducibility, standardization of cancer organoid formation, mimic the specific composition and structure of the native ECM, and high-throughput screening. Primary human glioblastoma (GBM) organoids are encapsulated in a hybrid material comprising synthetic polyethylene glycol (PEG) decorated with the RGD integrin-binding peptide and crosslinked with recombinant hyaluronic acid (HA) [88]. PEG-based dynamic hydrogels functionalized with the basement membrane protein laminin enable reproducible intestinal stem cell (ISC) expansion and organoid formation [89].

Engineered matrices for organoid cultures can optimize specific organoid systems, including stiffness, matrix viscoelasticity, and degradability. Organoids based on purified silk protein and alginate polysaccharides enable ECM systems to retain their superior homogeneity and reproducibility [90,91]. Fibrin matrices provide suitable physical support and naturally occurring Arg-Gly-Asp (RGD) adhesion domains on the scaffold, as well as supplementation with laminin-111; these are key parameters required for robust organoid formation and expansion [92]. A hybrid matrix comprising HA and elastin-like protein (ELP) regulate the growth rate of intestinal organoids and their formation efficiency [93]. A synthetic hydrogel extracellular matrix was designed for pancreatic organoids and replicated the phenotypic traits characteristic of the tumor environment in vitro [94].

### 2.3. Tumor Microenvironment of Organoids

#### 2.3.1. Culture Medium

Organoid media are necessary components in the establishment of cancer microenvironments in vitro. For example, Wnt 3a, R-spondin, Noggin, and epidermal growth factor are not dispensable for organoid growth. To preserve organoid heterogeneity, the omponents of the organoid media must be very complex. Wnt protein is at the heart of organoid technology. According to researchers who identify and develop the driver of TME, intestinal organoids are strictly dependent on Wnt ligands for survival and growth [95]. A set of growth factors are critical components of organoid media, which include R-spondins and BMP signaling antagonists such as Noggin or Gremlin 1 [96]. A B27 supplement comprises various nutrients, which increase cell survival rate, promote the formation of tumor spheres, and prevent sphere adhesion during lung cancer organoid culturing [97]. N-acetylcysteine is an antioxidant directly scavenging ROS and partially via ERK1/2 activation, and also a precursor of antioxidant glutathione (GSH) [98]. FGF7 and FGF10 are mainly used to induce lung organoid branching [99]. EGF belongs to a family of growth factors that drive proliferation in different organoids, such as colon cancer organoids [23], breast cancer organoids [69], pancreatic cancer organoids [24], and prostate cancer organoids [53]. A83-01 is a TGF-β/activin receptor-like kinase 5 (ALK5) inhibitor. The overexpression of TGF-β induces epithelial-to-mesenchymal transition (EMT) and facilitates immunosuppression, ECM deposition, and angiogenesis. Therefore, A83-01 inhibits the EMT of cancer organoids by inhibiting TGF-β [26]. In addition, other factors play different roles in organoid culturing, such as the N2 supplement, nicotinamide, prostaglandin E2, and gastrin 1, which induce differentiation, promote cell survival, and so on [100].

#### 2.3.2. Non-Neoplastic Cells

The co-culturing of primary tumor epithelia with endogenous, syngeneic tumor-infiltrating lymphocytes (TILs), which comprise patient-derived organoids (PDOs), enables the successful modeling of the immune checkpoint blockade (ICB) with anti-PD-1- and/or anti-PD-L1, expanding and activating tumor antigen-specific TILs and eliciting tumor cytotoxicity [101]. Co-culturing with cancer-associated fibroblasts (CAFs) enhances the organoid-forming ability of CD44+ cells, as well as the expression of CD44 and OCT-4 at the protein level [102]. Multi-cell type organotypic co-culture models include stromal cells, immune cellular components, and pancreatic cancer organoids. The activation of myofibroblast-like cancer-associated fibroblasts and tumor-dependent lymphocyte infiltration were observed in these models [103]. The microenvironment includes CAFs, tumor endothelial cells (TECs), tumor-associated macrophages (TAMs), and regulatory T cells (Tregs), which influence and recapitulate cancer progression [104,105].

#### 2.3.3. Angiogenesis

Primary HUVECs were co-cultured with primary colorectal tumoroids to observe induced angiogenesis via tube formation assay. The assay evidenced that tube formation increased in a dose-dependent manner upon treatment with the pro-angiogenic factor vascular endothelial growth factor A(VEGF-A) [106]. Co-cultures of hepatocellular carcinoma (HCC) cell lines or patient-derived xenograft organoids with endothelial cells exhibited the upregulation of MCP-1, IL-8, and CXCL16, influencing tumor necrosis factor (TNF) and angiocrine signaling, which generate an inflammatory microenvironment by recruiting immune cells [107]. Interactions with vasculature were found in spheroids and organoids upon 7 days of co-culture with space of Disse-like architecture in between hepatocytes and endothelium, resulting in a stable, perfusable vascular network [108]. Vascularized spheroids were generated from a non-adherent microwell culture system of human umbilical vein endothelial cells, human dermal fibroblasts, and human umbilical-cord-blood-derived mesenchymal stem cells to develop and assemble vascular spheroids with cerebral organoids [109]. The microvascular network cultured with patient-derived tumor organoids presented highly angiogenic features. ECM components and the culture media composition were adjusted to coculture patient-derived colon organoids, which form a self-assembled microvasculature under intravascular perfusion by using a microfluidic platform [110]. The relations between platelets and cancer organoids are also prospective for tumor angiogenesis studies [111,112].

## 3. A Graphene Oxide Platform for Cancer Research

### 3.1. Properties of Graphene Oxide

Graphene, a two-dimensional honeycomb lattice of carbon atoms, is typically synthesized via chemical vapor deposition or mechanical exfoliation of graphite, noted for its exceptional electrical, mechanical, and thermal properties [113]. GO created using the Hummers method introduces oxygen-containing functional groups to graphite, resulting in a material with a large surface area and functional versatility, albeit with reduced electrical conductivity compared with graphene [114,115]. Reduced graphene oxide (r-GO), obtained by removing these oxygen groups from GO, restores some of graphene’s electrical and structural features [116]. Graphene quantum dots (GQDs), small graphene fragments synthesized through top–down or bottom–up methods, exhibit unique optical and electronic properties due to quantum confinement and edge effects [117]. Both GO and r-GO, rich in functional groups, are adaptable for diverse modifications and applications, particularly in biosensors and environmental remediation, while GQDs find use in fluorescence-based sensors and electronics [118]. Graphene’s integration into composites boosts their mechanical, thermal, and electrical characteristics, with r-GO also being leveraged in similar applications for its balance between conductivity and functional compatibility [119]. We discuss the detailed properties of GO in the following sections.

#### 3.1.1. Mechanical Properties

GO consists of stacked layers of graphene sheets that are held together by oxygen-containing functional groups, creating a layered, lamellar structure. The structure of graphene is a monolayer of two-dimensional (2D) one-atom-thick planar, sp2-hybridized carbon arranged in six-atom rings [120]. GO is a complex material due to its amorphous and non-stoichiometric atomic composition. Currently, there are no precise analytical techniques available for the thorough characterization of GO materials and their analogs. According to previous research, several structural models have been proposed, including models by Hofmann, Ruess, Scholz-Boehm, Nakajima-Matsuo, Lerf-Klinowski, and Dekany. In terms of regular lattices, they consist of discrete repeated units, and the Lerf-Klinowski model is currently considered the most widely accepted configuration [121,122]. Brittle fracture ensues when the material is subjected to a critical stress, typically around 130 GPa, in accordance with its intrinsic strength. Because of its high values of E (elastic modulus) and σint (intrinsic strength), graphene is regarded as an exceptionally robust material for structural applications. Graphene oxide paper, on average, exhibits a fracture strength of 80 MPa and an elastic modulus of 32 GPa [123].

#### 3.1.2. Water Dispersibility

GO is highly hydrophilic and readily disperses in water and other polar solvents due to the presence of hydrophilic functional groups. This property is advantageous for various applications, such as nanocomposites and biomedical applications. The presence of oxygen-containing functional groups such as -OH, -COOH, and epoxide groups on the surface makes GO hydrophilic. Félix Mouhat et al. evidenced that GO is chemically active in water, acquiring an average negative charge of the order of 10 mCm^−2^ [124]. The hydrophilicity of graphene oxide with different particle sizes and pH values was characterized by the water contact angle. And Xuebing Hu et al. found that the water contact angle of the different graphene oxides decreased from 61.8° to 11.6°, which indicates graphene oxide has excellent hydrophilicity [125].

#### 3.1.3. Thermal Properties

Graphene stands out as an exemplary thermal conductor, demonstrating an impressive thermal conductivity range of 2000–5000 W/mK [126]. Nevertheless, the introduction of oxygen functional groups onto the surface of graphene oxide (GO) disrupts the lattice symmetry and induces localized strain, resulting in a substantial reduction in thermal conductivity by orders of magnitude (2–3 orders of magnitude, to be precise) [127]. A remedy for this attenuation in thermal conductivity involves the partial reduction of GO through chemical reactions with reducing agents, notably hydrazine and its derivatives. This process culminates in the creation of reduced graphene oxide (rGO) by effectively extracting oxygen functional groups. An interesting observation has been made regarding the thermal conductivity of GO, which exhibits a continuous decrease as the degree of oxidation escalates [128]. Studies show the successful formation of GO/RGO–protein complexes with enhancement in structural/thermal stability due to various interactions at the nano–bio interface and their utilization in various functional applications [129].

#### 3.1.4. Electrical Properties

Graphene oxide is inherently non-conductive, necessitating the removal of a significant portion of its oxygen groups for conversion into reduced graphene oxide (rGO) to enhance its electrical conductivity. Researchers have observed a distinct difference in electric conductivity between R-I-Ph-GO/PI films and R-GO/PI films, primarily attributed to the formation of a sp2-hybrid carbon network within the graphene oxide structure [130]. Notably, the electrical properties of GO films exhibit sensitivity to both humidity levels and applied voltage amplitude [131]. At low humidity, GO films demonstrate poor conductivity, akin to insulators. Conversely, under high humidity conditions, GO film conductivity markedly increases because of enhanced ion conduction mechanisms, offering insights into tailored electrical properties for GO-based materials in applications influenced by environmental factors, particularly humidity.

#### 3.1.5. Chemical Properties

GO offers superior dispersibility in various mediums, including water, diverse solid matrices, and organic solvents. This property makes it highly versatile. GO can be effectively combined with polymer or ceramic matrices to form composite materials, often resulting in improved electrical and mechanical properties [121]. However, GO does come with certain limitations, such as the potential for agglomeration or overlapping of GO sheets. Nonetheless, the thin and flat structure of GO sheets allows for flexibility in making structural and morphological modifications. This flexibility is further enhanced by the presence of oxygen functional groups in GO, which serve both as sites for functionalization and as spacers for molecular absorption. These attributes facilitate the incorporation of GO into various nano composites and nano-morphologies. By employing structural modifications and functionalization through covalent and noncovalent bonding interactions, GO finds applications in a wide array of real-world uses, including filtration membranes [132], electrochemical sensors [133], hydrogen storage devices [134], battery electrodes [135], supercapacitors [136,137], and microjet engines [138].

#### 3.1.6. Optical Properties

GO has a range of remarkable optical properties driven by its electronic configuration. These properties include structure-dependent absorption and Raman spectra, which provide insights into its chemical composition and the extent of functionalization-induced disorder [139]. In contrast to pristine graphene, GO displays photoluminescence across a spectrum encompassing ultraviolet, visible, and near-infrared wavelengths, contingent upon its structural variations. Reduced graphene oxide (rGO), on the other hand, exhibits the capacity to absorb radiation across an extensive wavelength range spanning from ultraviolet to terahertz frequencies [140]. To investigate the effects of modifications on fluorescence behavior, various agents, such as polyethylene glycol (PEG) polymers, metal nanoparticles (including Au and Fe_3_O_4_), and folic acid (FA) molecules, have been employed to functionalize the surface of GO [141]. rGO was functionalized with L-arginine (L-Arg) that on the optically active support generated an effective optical chemosensor for the determination of Cd (II), Co (II), Pb (II), and Cu (II) [133]. The marriage between integrated photonics and GO has led to the birth of integrated GO photonics, which has become a very active and fast-growing branch of on-chip integration of 2D materials in order to achieve novel functionality of integrated photonic devices [140].

#### 3.1.7. PH-Sensitivity

Graphene oxide-based nanomaterials can be affected by pH changes in their surface properties. The tumor environment is generally more acidic (pH 6.4~7) than normal cells (pH 7.4) [142,143]. A dual-targeting drug delivery and pH-sensitive controlled release system GO–Fe_3_O_4_ nanohybrid has been established [144]. β-cyclodextrin grafted L-phenylalanine functionalized graphene oxide is a versatile nanocarrier for pH-sensitive doxorubicin delivery [145]. GO was functionalized covalently with pH-sensitive poly(2-(diethylamino) ethyl methacrylate) (PDEA) by surface-initiated in situ atom transfer radical polymerization. Simple physisorption by π-π stacking and hydrophobic interactions on GO-PDEA can be used to load camptothecin (CPT), a water-insoluble cancer drug that is released only at lower pH levels normally found in a tumor environment but not in basic and neutral pH circumstances [146]. The constructed graphene oxide hybrid cyclodextrin-based supramolecular hydrogels could respond to NIR light, temperature, and pH, which could be beneficial for the controlled release of cargoes. Graphene oxide sheets not only acted as a core material to provide additional cross-linking but also absorbed NIR light and converted NIR light into heat to trigger the –sol–gel transition [147].

### 3.2. Graphene Oxide in Cancer Diagnosis

The most effective approach for curbing the advancement of cancer is the development of innovative diagnostic tools that enable the detection of the disease at its early stages. The early detection of carcinoma cells and the continuous monitoring of their activities hold significant importance in the realms of clinical diagnostics, toxicity monitoring, and safeguarding public health. Detecting and monitoring tumor cells during their early stages play a pivotal role in preventing the progression of cancer. This is particularly crucial in cases such as pancreatic cancer, which is challenging to diagnose at an early stage and often leaves limited possibilities for rescuing patients in advanced stages of the disease. Biomedical imaging technologies serve as highly efficient tools for tumor diagnosis, providing invaluable insights that can effectively guide tumor therapeutics. GO serves as a versatile agent for bioimaging functions due to its unique physical and chemical properties (Figure 2).

#### 3.2.1. Magnetic Resonance Imaging

Magnetic resonance imaging (MRI) stands out as a non-invasive and non-ionizing diagnostic method, renowned for its exceptional spatial and temporal resolution. In the context of MRI, nanomaterials based on GO functionalized with paramagnetic metals have demonstrated great promise. Here, dendrimers featuring amino group caps (DEN) are skillfully grafted onto GO nanosheets which lateral sizes in the range of 40–380 nm (mean size ∼175 nm). This grafting process serves as a crucial step, enabling the subsequent functionalization of GO with gadolinium diethylene triamine pentaacetate (Gd-DTPA) and prostate stem cell antigen (PSCA) monoclonal antibody (mAb). Remarkably, the in vivo results obtained from magnetic resonance imaging validate the utility of GO-DEN(Gd-DTPA)-mAb as a targeted contrast agent for prostate tumor imaging [148]. Scientists have integrated GO with superparamagnetic iron oxide nanoparticles (Fe_3_O_4_ NPs). These Fe_3_O_4_ NPs serve a dual purpose, functioning as both biocompatible magnetic drug delivery enhancers and magnetic resonance contrast agents for MRI. The synthesized GO-Fe_3_O_4_ conjugates exhibit an average size of 260 nm and demonstrate low cytotoxicity levels, which are on par with those observed in GO alone [149]. Researchers focused on the synthesis of hybrid nanocomposites, specifically graphene oxide–zinc ferrite (GO-ZnFe_2_O_4_), which are further conjugated with doxorubicin (DOX) for applications in cancer therapy and MRI-based diagnosis. GO-ZnFe_2_O_4_ and GO-ZnFe_2_O_4_/DOX ranging from 5 to 100 μg/mL were investigated, and the key components are zinc ferrite (ZnFe_2_O_4_) nanoparticles (NPs), which serve as MR imaging contrast agents [150].

The size of magnetic nanoparticles (MNPs) in graphene oxide (GO) composites critically influences their performance, with smaller MNPs enhancing surface reactivity, ensuring superparamagnetic behavior, and improving biological penetration and distribution, while also affecting stability and toxicity profiles [151]. This size-dependent variation in physical and magnetic properties is fundamental in tailoring GO-MNP conjugates for specific applications, particularly in biomedical fields such as MRI [152].

#### 3.2.2. Fluorescence Imaging

Fluorescence lifetime imaging (FLIM) is a non-invasive technique reliant on photons emitted by fluorescent probes, frequently employed for monitoring pathological tissue. A graphene oxide–MnO_2_–fluorescein (GO–MnO_2_–FL) nanocomposite was synthesized and applied for the detection of reduced glutathione (GSH). GSH, an essential endogenous antioxidant, plays a pivotal role in cellular defense against reactive oxygen species (ROS), thereby maintaining cellular activity. Notably, the GO–MnO_2_–FL (100 μg/mL) nanocomposite exhibited selective imaging of cancer cells, owing to the higher GSH content observed in cancer cells as compared with normal cells [153]. A non-invasive and targeted technology for early diagnosis of oral squamous cell carcinoma (OSCC) involves the synthesis of nano-graphene oxide (NGO) nanoparticles. These nanoparticles are designed with specificity to the gastrin-releasing peptide receptor (GRPR), achieved through the incorporation of GRPR-specific peptides AF750-6Ahx-Sta-BBN. This approach enables precise and non-invasive near-infrared fluorescence imaging targeted at OSCC [154]. GO-based fluorescent DNA nanomaterials offer a promising avenue for the in vitro diagnosis and therapy of liver tumor cells [155].

#### 3.2.3. Photoacoustic Imaging

Photoacoustic Imaging (PAI) stands as a robust diagnostic tool hinging on the photoacoustic (PA) effect. The distinctive capability of PAI lies in its ability to furnish optical absorption contrast and achieve high-resolution imaging, rendering it particularly well-suited for applications involving deep tissue and organ imaging [156]. PA imaging is a noninvasive imaging modality that depends on the light absorption coefficient of the imaged tissue and the injected PA-imaging contrast agents. PA imaging integrates the excellent contrast achieved in optical biomedical imaging with the deep penetrability of ultrasound (US) imaging. Thus, PA imaging can be used for the imaging of deeper tissue compared to other optical imaging technologies [157].

A nanotheranostic agent has been fabricated by direct deposition of Bi_2_Se_3_ nanoparticles on graphene oxide (GO) in the presence of polyvinylpyrrolidone (PVP) using a one-pot solvothermal method. GO/Bi_2_Se_3_/PVP nanocomposites (2 mg/mL) could serve as an efficient bimodal contrast agent to simultaneously enhance X-ray computed tomography imaging and photoacoustic imaging in vitro [158]. The reduced graphene oxide coated gold nanorods (r-GO-AuNRs) and highly efficient heat transfer process through the reduced graphene oxide layer, r-GO-AuNRs (0.125 mg/mL), exhibit excellent photothermal stability and significantly higher photoacoustic amplitudes; therefore, r-GO-AuNRs can be a useful imaging probe for highly sensitive photoacoustic images [159]. A sandwich-type gold nanoparticle coated reduced graphene oxide (rGO-AuNP) as an effective nanotheranostic platform for the second near-infrared (NIR-II) window photoacoustic (PA) imaging-guided photothermal therapy (PTT) in ovarian cancer [160]. With nanoparticles composed of a liquid gallium core with a reduced graphene oxide (RGO) shell (Ga@RGO) of tunable thickness, the high near-infrared absorption of RGO results in a photothermal energy conversion of light to heat of 42.4%. This efficient photothermal conversion, combined with the large intrinsic thermal expansion coefficient of liquid gallium, allows the particles to be used for photoacoustic imaging, that is, the conversion of light into vibrations that are useful for imaging [161].

Indocyanine green (ICG)-loaded, polyethylene glycosylated (PEG), reduced nano-graphene oxide nanocomposite (rNGO-PEG/ICG) is a new type of fluorescence and photoacoustic dual-modality imaging contrast. The nanocomposite is demonstrated to possess greater stability, longer blood circulation time, and superior passive tumor targeting capability, which can be a promising candidate for further translational studies on both the early diagnosis and image-guided therapy/surgery of cancer [162].

#### 3.2.4. Raman Imaging

Raman scattering (SERS) is widely used due to its non-invasiveness, ultrasensitivity, and high spatial resolution. Since molecular vibrations are strongly related to the molecular structure, condition, and environment, the combination of spontaneous Raman scattering can be used to monitor both the location and condition of biological molecules in living cells [163]. GO possesses characteristic fingerprints in Raman spectra; therefore, it is also used for Raman imaging with Au and Ag nanoparticles loaded as SERS substrates [164]. GO/gold nanoparticle (AuNP) hybrids with folic acid (FA) binding are prepared as a multifunctional platform in bioimaging. FA is the targeting agent, AuNPs work as surface-enhanced Raman scattering substrates, and GO takes the role of both supporting the AuNPs with FA and acting as a Raman probe [165]. Methylene blue-loaded mesoporous silica-coated gold nanorods on graphene oxide (MB-GNR@mSiO_2_-GO) (36 µg/mL) were developed as an all-in-one photo-nanotheranostic agent for intracellular surface-enhanced Raman scattering (SERS) imaging-guided photothermal therapy (PTT)/photodynamic therapy (PDT) for cancer [166].

Afua A. Antwi-Boasiako and colleagues reported the use of bioconjugated 2D graphene oxide (bio-GO) nanostructures, with the average lateral size of the layered GO sheets being approximately 0.08–0.1 μm. These were used as probes for breast cancer cells (SKBR3), demonstrating excellent discrimination over other types of circulating tumor cells by monitoring the ‘turn-off’ of the Raman signal [167]. Lin Yang et al. designed silver nanoparticles deposited on graphene oxide for ultrasensitive surface-enhanced Raman scattering immunoassay of cancer biomarkers, which made the detection of prostate-specific antigen (PSA) serum samples from prostate cancer patients satisfactory and thereby demonstrating that sensitive enzyme-assisted dissolved AgNPs SERS immunoassays of PSA have potential applications in clinical diagnosis [168].

#### 3.2.5. Computed Tomograph

Computed tomography (CT) is a widely adopted disease diagnosis method in clinical settings because of its non-invasiveness and high spatial resolution properties, which are based on its high atomic number and X-ray absorption. Graphene oxide/gold nanorod (GO/GNR) nanohybrids were synthesized with a GO- and gold-seed-mediated in situ growth method. Upon injection of the GO/GNRs (50 μg/mL) into xenograft tumors, excellent CT imaging properties and photothermal effect were obtained [169]. Zhan Li et al. evidenced that Ag nanoparticles (AgNPs) are composited on the surface of GO to promote its X-ray absorption, and then simvastatin is coinjected in mice of renal dysfunction to eliminate in vivo toxicity-induced by AgNPs [170]. One new composite contrast agent based on Ln and graphene matrices was developed for multi-energy computed tomography [171].

Graphene nanoplatelets (GNPs), synthesized via potassium permanganate-based oxidation and exfoliation followed by reduction with hydroiodic acid (rGNP–HI), intercalated with manganese ions within the graphene sheets, and covalently functionalized with iodine, exhibit excellent potential as bimodal contrast agents for magnetic resonance imaging (MRI) and CT [172]. A novel system for synergistic cancer therapy was developed based on bismuth sulfide (Bi_2_S_3_) nanoparticle-decorated graphene functionalized with polyvinylpyrrolidone (PVP) (named PVP-rGO/Bi_2_S_3_). GO nanosheets with an average diameter of ~100 nm and a thickness of ~1.2 nm were prepared via a modified Hummer’s method. Due to the obvious NIR and X-ray absorption ability, the PVP-rGO/Bi_2_S_3_ nanocomposite could be employed as a dual-modal contrast agent for both photoacoustic tomography and X-ray computed tomography imaging [173]. By using a solvothermal method in the presence of polyethylene glycol (PEG), BaGdF_5_ nanoparticles are firmly attached to the surface of GO nanosheets to form the GO/BaGdF_5_/PEG nanocomposites, thus enabling effective dual-modality MR and X-ray computed tomography (CT) imaging of the tumor model in vivo and indicating the potential applications of dual-modality MR/CT imaging of cancers [174].

### 3.3. Graphene Oxide in Cancer Treatment

#### 3.3.1. Delivery System

Graphene-based materials have high specific surface areas and low toxicity. GO-based nanomaterials have been extensively studied in cancer treatment (Figure 2), for instance, as a new type of nanocarriers to deliver various therapeutic anticancer agents. In this section, we discuss recent advancements in GO delivery systems for cancer research. Notably, we compile a list of significant recent studies on functionalized GO delivery systems (refer to Table 1). This list showcases the progress and innovations in the application of GO for targeted delivery systems of cancer therapy.

##### Drug Delivery

GO can be functionalized with polymers. Chitosan (CS) is an amino polysaccharide, a hydrophilic, biocompatible, non-toxic, and biodegradable polymer of glucosamine and acetylglucosamine [175]. To synthesis GO-CS is involved in amide coupling between the COOH group on the GO and the amino group of CS. CS improves the solubility of GO sheets in acidic media. Moreover, GO-CS results in changes in particle size and zeta potential as a function of pH [176,177]. Yan et al. prepared GO-CS and investigated its potential as a nanoadjuvant. GO-CS (50 μg/mL) significantly activated RAW264.7 cells and stimulated more cytokines for mediating cellular immune responses [178].

PEG is a polymer of repeating ethylene ether units that is widely used for several pharmaceutical and biomedical applications. It is soluble in aqueous and organic media, being defined as biocompatible, biodegradable, non-toxic, non-immunogenic, and is classified as “Generally Regarded as Safe” (GRAS) by the FDA [179]. Nano-graphene oxide (NGO) was synthesized its biological applications were explored in order to develop functionalization chemistry to impart solubility and compatibility to NGO in buffer solutions and other biological media by covalently grafting PEG star polymers onto its chemically activated surfaces and edges [180]. To evaluate it as a potential anti-metastatic agent, GO was modified with polyethylene glycol to form PEG-modified GO (PEG-GO). PEG-GO did not show apparent effects on the viability of breast cancer cells (MDA-MB-231, MDA-MB-436, and SK-BR-3) or non-cancerous cells (MCF-10A), but inhibited cancer cell migration in vitro and in vivo. An analysis of cellular energy metabolism revealed that PEG-GO significantly impaired mitochondrial oxidative phosphorylation (OXPHOS) in breast cancer cells but not in non-cancerous cells [181]. A de novo drug delivery nanosystem (~128 nm) based on gold nanoparticles (GNPs), decorated PEG, and folate (FA)-conjugated GO was designed to load with doxorubicin hydrochloride (DOX) as a model anticancer drug. Its drug-loading capacity as well as pH-dependent drug release behavior were investigated [182].

Hyaluronic acid (HA) is a naturally mucopolysaccharide, biocompatible, non-immunogenic, and biodegradable polysaccharide, consisting of alternating units of D-glucuronic acid and N-acetyl-D-glucosamine [183]. Numerous tumor cells overexpress several receptors that have a high binding affinity for HA, while these receptors are poorly expressed in normal body cells. Graphene quantum dots (GQD) were used as drug carriers, and HA was decorated on the surface of GQD to target cancer cells. At the same time, curcumin (CUR) was used as a drug model and loaded on the synthesized nanocarriers. GQD-HA-CUR reduces HeLa cell viability significantly because of the mediation of HA–CD44 for drug cell uptake [184]. Metformin was loaded upon GO via drop-wise addition of 2 mL (10 mg/mL) of metformin into 20 mL of GO dispersion (5 mg/mL). HA was grafted onto metformin-loaded GO nanoparticles as a CD44-targeted anti-cancer therapy for triple-negative breast cancer, which exhibited anti-cancer efficacy at a much lower dosage as compared with metformin alone [185]. Doxorubicin (Dox) and paclitaxel (Ptx) were successfully loaded onto GO-HA that covalently attached HA onto GO. The GO-HA-Dox/Ptx system was significantly better than the GO-Dox/Ptx system at specifically killing CD44-expressing MDA-MB-231 cells but not BT-474 cells without the expression of CD44 [186].

Polyvinyl alcohol (PVA) is a water-soluble polymer synthesized via hydrolysis and radical polymerization of vinylacetate [187,188]. Curcumin was loaded in PVA-sodium alginate/3D-GO hydrogels for studying in vitro drug delivery systems [189]. Magnetic magnesium iron oxide nanoparticles were synthesized via the coprecipitation chemical method and then composited with graphene oxide and modified by polyvinyl alcohol. Paclitaxel (PTX) and docetaxel (DTX) were loaded in the modified magnetic nanocomposites. The generally sustained and controlled release profile of DTX (or PTX) facilitates the application of modified nanocomposite for the delivery of anticancer drugs [190].

Polyacrylic acid (PAA) is a synthetic polymer of acrylic acid monomers, which is biocompatible, non-toxic, pH-sensitive, and mucoadhesive. In aqueous solutions, PAA has an anionic nature because of its carboxylic groups [191]. Nanocomposite systems, consisting of a reduced graphene oxide/polyacrylic acid as a nanocarrier, were integrated with a folic-acid-targeting agent and further modified by Deferrioxamine-M (M: Mn^2+^ or Gd^3+^) as the diagnostic MRI contrast agent or Temozolomide as the therapeutic agent. Release studies at a biological of pH 7.4 revealed good stability for TMZ immobilized on the GNs@PAA-FOA/TMZ nanocarrier [192]. Gemcitabine (GEM)—PAA—GO are developed in an explicit solvent medium at two different pH values, which can control drug biodistribution in response to changes in pH that are markers of the tumor environment [193].

Polyvinylpyrrolidone (PVP) is a water-soluble, non-ionic, non-toxic polymer surfactant. The coating of PVP is reported to improve the dispersibility and biocompatibility of GO in physiological buffers. Injectable hydrogel polymeric nanoparticles of PVP cross-linked with N, N′-methylene bis-acrylamide and encapsulating water-soluble macromolecules FITC–dextran (FITC–Dex) have been prepared in the aqueous cores of reverse micellar droplets, which serve as a potential carrier for hydrophilic drugs [194]. A stimuli-responsive polyvinylpyrrolidone-NIPPAm-lysine graphene oxide nano-hybrid was designed and fabricated for the delivery of chemotherapeutic agent fluorouracil (FU) to MCF7 breast cancer cells [195]. Nanocarriers comprising gelatin (G)-PVP coated GO were prepared and loaded with quercetin (QC). Additionally, a dual nanoemulsion water/oil/water with bitter almond oil was developed as a membrane around the nanocomposite to control further drug release. The pH-sensitive drug delivery system showed an 87.5% encapsulation efficiency and a 45% drug loading, which are among the highest values reported up to date. MTT assay and flow cytometry methods revealed a rate of cancer cell death of 53.14%, which was 36.51% in the apoptotic phase [196].

Dextran (Dex) is a hydrophilic natural polymer and a polysaccharide synthesized from the condensation of glucose. Cellular experiments uncover that DEX coating on GO offers remarkably reduced cell toxicity [197]. The non-covalent functionalization of GO with chitosan (CS) and Dex was successfully developed via a layer-by-layer self-assembly technique for anti-cancer drug delivery application. The CS/Dex functionalized GO nanocomposites (GO-CS/Dex) exhibited a diameter of about 300 nm and a thickness of 60 nm and showed a strong cytotoxicity to the cancer cells [198].

Graphene oxide/cationic polyethyleneimine/poly anionic dextran sulfate (GO/PEI/DS) was synthesized via a layer-by-layer self-assembly technique for transdermal anti-cancer drug delivery. DOX was loaded onto folic-acid-conjugated GO and methotrexate (MTX) was loaded onto dextran sulfate (DS). The results revealed that the synthesized dual drug-loaded material showed a good pH-dependent controlled and sustained release profile for both DOX and MTX via the transdermal route of administration in comparison with oral delivery [199]. GO-PEI complexes were loaded with a miR-214 inhibitor which efficiently inhibited cellular miR-214, resulting in a decrease in oral squamous cell carcinoma (OSCC) cell invasion and migration and an increase in cell apoptosis [200].

##### Gene Delivery

Gene therapy needs a vector that can protect genes from nuclease degradation and facilitate gene uptake with high transfection efficiency. Compared with viral vectors, non-viral gene carriers can avoid immune responses, toxicity, chromosomal integration, and so on. GO-based vectors for gene delivery have been considered as superior vehicles because of their good biocompatibility, biosafety, large surface area, adsorption capacity, and negative charges [201].

Polyethyleneimine (PEI) is a water-soluble cationic polymer with a large number of amino groups. GO was modified with PEG, PEI, and FA for the targeted delivery of small-interfering RNA (siRNA) that inhibits ovarian cancer cell growth [202,203]. Lactosylated chitosan oligosaccharide (LCO)-functionalized GO (GO-LCO) containing quaternary ammonium groups (GO-LCO^+^) was prepared to realize the hepatocyte-specific targeted delivery of anti-tumor drugs and genes. The GO-LCO^+^ could be used to load Dox with the loading efficiency of 477 μg/mg and fluorescein FAM-labeled DNA with 4 μmol/g, respectively. Results suggest that the functionalized GO can be used as a nanocarrier for the hepatocyte-targeted co-delivery of anti-tumor drugs and genes with low cytotoxicity, indicating its potential future applications in anticancer drug-and-gene combined therapy [204].

Glycyrrhetinic acid (GA) is employed as a liver-targeting ligand to construct GA, polyethylene glycol (PEG), polyamidoamine dendrimer (Dendrimer), and nano-graphene oxide (NGO) conjugates (GA-PEG-NGO-Dendrimer, GPND) for siRNA delivery. The GPND/siRNA nanocomplex has high safety, targeting, and transfection as well as a prolonged half-life [205]. A modified GO nanocarrier for the co-delivery of siRNA and DOX was designed for enhanced cancer therapy. GO-poly-l-lysine hydrobromide/folic acid (GPF)/DOX/siRNA exhibited gene silencing and tumor inhibition [206]. GO/3-aminopropyltriethoxysilane (APTES) was modified via spermine (GOAS), which acts as a gene delivery system to help the transfection of pEGFP-p53 into breast cancer cell lines [207]. Nanoparticles comprising GO/cationic lipid (GOCL) condense and stabilize plasmid DNA for transfection into human cervical cancer (HeLa) cells [208].

##### Antibody Delivery

A pH-responsive charge-reversal polyelectrolyte and integrin αⅤβ3 mono-antibody functionalized GO complex was developed as a nanocarrier for the targeted delivery and controlled release of DOX into cancer cells [143]. A GO platform was functionalized with magnetic nanoparticles and a monoclonal antibody specific to the carbonic anhydrase marker. CA IX is a cell surface hypoxia-inducible enzyme functionally involved in adaptation to acidosis that is expressed in aggressive tumors [209]. Multifunctional mesoporous silica nanoparticles (MSNs) have internal fluorescent conjugates and external polydopamine and GO layers. Monoclonal antibody (anti-human epidermal growth factor receptor)-conjugated MSNs showed a higher specificity, which resulted in more enhanced anticancer effects in vitro [210].

GO was modified with a non-toxicity cationic material (chitosan) and a tumor-specific monoclonal antibody (anti-EpCAM) for the delivery of survivin-siRNA (GCE/siRNA). It was demonstrated that GCE/siRNA had a strong antitumor effect in vitro [211]. An antibody-modified reduced graphene oxide (rGO) film efficiently captured circulating tumor cells (CTCs) and minimized the background of white blood cells without complex microfluidic operations [212]. The noncovalent association of anti-HER2 antibody trastuzumab (TRA) with GO generates stable TRA/GO complexes that are capable of rapidly killing osteosarcoma (OS) cells [213].

A single layer of carboxymethylcellulose (CMC) and poly N-vinylpyrrolidone (PVP) was cross-linked through a disulfide bond and deposited on graphene oxide nanoparticles (GO NPs). The NPs were functionalized via monoclonal antibody FA, which showed a high inhibition of Saos2 and MCF7 cell lines in vitro [214]. Yang et al. evidenced the efficient targeting of breast cancer metastasis in an experimental murine model featuring GO conjugated with a monoclonal antibody (mAb) against follicle-stimulating hormone receptor (FSHR), a highly selective tumor vasculature marker in both primary and metastatic tumors [215].

**Table 1 ijms-25-01066-t001:** Delivery systems in recent studies on functionalized GO.

Type of Nanomaterials	Advantages	Limitations	Carriers (Drugs/Gene/Antibody)	Reference
GO-CS-(FA)	smaller size, positive surface charge, increased compatibility, high loading capacity	thermally unstable	drug: Chitosan/Folic acid	[176]
GO–PEG–FA/GNPs–DOX	high drug loading capacity, acceptable biocompatibility, pH-responsive drug release profile	slow drug release	drug: Doxorubicin hydrochloride	[182]
HA-GO-Met	selectively targeting CD44+ triple-negative breast cancer (TNBC) cells, anti-cancer efficacy in vitro and in vivo	complex synthetic method	drug: Metformin	[185]
PVA-SA/3D-GO	lower swelling and higher release	complex synthetic method, percentages of GO influence delivery	drug: Curcumin	[189]
(DTX/PTX)-MgFe_2_O_4_-GO-PVA	sustained and controlled release	complex synthetic method without an in vivo test	drug: Paclitaxel, Docetaxel	[190]
FU-GO/NHs	FU-GO/NHs more cytotoxic than free FU, fast uptake, and temperature and pH-responsive	no normal cells for comparison	drug: 5-fluorouracil	[195]
GO/PEI/DS-DOX-MTX	dual drug loading, good selectivity, prolonged drug existence in the blood circulation system, and pH-dependent controlled and sustained release	complex synthetic method	drug: Doxorubicin, Methotrexate	[199]
PEG-GO-PEI-FA/siRNA	siRNA condensation and stable	complex synthetic method without an in vivo test	gene: siRNA	[203]
GO-LCO^+^(-FAM-DNA)	co-delivery of anti-tumor drugs and genes with low cytotoxicity	complex synthetic method	drug: Doxorubicin chloride gene: DNA	[204]
GA-PEG-NGO-Dendrimer/anti-VEGFa siRNA	stability, low toxicity, negligible hemolytic activity, high transfection efficiency, and targeted	complex synthetic method	gene: anti-VEGFa siRNA	[205]
APTES-GO-(pEFGP-*p53*)	Ninety percent of the cells were transfected	unstable during the heating process, complex synthetic method, and no in vivo test	gene: EFGP-p53 plasmid	[207]
GOCL/DNA	high transfection, low toxicity, a library of 9 cationic lipids screening	no biological milieu test	gene: DNA	[208]
GO-Abs/PEI/PAH-Cit/DOX	targeted cancer cells, release by mild acidic pH stimulation	complex synthetic method	antibody: integrin αⅤβ3 drug: Doxorubicin	[143]
GO-MNps-EDC-MAb	targeted drug delivery, enhanced biocompatibility	potential cytotoxicity, stability, and uniformity of the composites	antibody: CA IX cDNA	[209]
MSNs-CPT@A-F@PDA@GO	MSNs exhibited stimuli (pH, NIR irradiation)-responsive controlled release, a higher specificity, and efficient cytotoxicity toward cancer cells	complex synthetic method, without an in vivo test	antibody: anti-human EGFR drug: Cisplatin	[210]
GO-CS-anti-EpCAM-siRNA	antitumor effect, accumulates siRNA in tumor tissues, and biosafety carrier	complex synthetic method, no significant differences between GCE/siRNA and Lipo/siRNA for downregulation rates of survivin-mRNA	antibody: anti-EpCAM gene: survivin-siRNA	[211]
CMC/PVP GO-FA-Curcumin	enhanced antiangiogenesis, apoptosis, and tumor growth inhibition	complex synthetic method	antibody: folic acid antibody drug: curcumin	[214]
NOTA-GO-FSHR-mAb	stability and high specificity for FSHR, use for early metastasis detection, and targeted delivery of therapeutics	complex synthetic method	antibody: anti-follicle-stimulating hormone receptor drug: Doxorubicin	[215]

#### 3.3.2. Phototherapy

Phototherapy, including photothermal therapy (PTT) and photodynamic therapy (PDT), is a newly developed and encouraging therapeutic strategy, which employs near-infrared (NIR) laser photoabsorbers to generate heat for thermal ablation of cancer cells upon NIR laser irradiation [216].

##### Photothermal Therapy

PTT operates by transforming radiant light energy into localized heat through external NIR laser irradiation. This process induces hyperthermia, raising the temperature at the tumor site and subsequently causing damage and apoptosis of tumor cells. Owing to its unique advantages of high specificity and minimal invasiveness, PTT has been evidenced as having great potential in treating cancer metastasis [216]. In contrast to conventional therapeutic methods such as chemotherapy, surgery, and radiotherapy, the utilization of NIR light within the 700–1100 nm range to induce hyperthermia holds particular appeal. This is because biological systems generally lack chromophores that absorb within the NIR region [217].

PTT necessitates a specific agent capable of being excited and transforming electromagnetic energy into heat upon exposure to a particular light source. Ideally, an ideal PTT agent should exhibit the following properties: (1) a high rate of photothermal conversion; (2) suitable biocompatibility; and (3) straightforward conjugation capacity. GO not only fully satisfies all of these requirements but also offers additional advantageous properties that are highly beneficial in the context of cancer PTT [218].

A biocompatible platform known as porphyrin-functionalized graphene oxide (PGO) has been synthesized and designed with a strong absorption capacity at 808 nm. This PGO platform, equipped with active functional groups, enables precise targeting in PTT for brain cancer treatment while preserving the well-being of healthy cells and tissues [219]. A theranostic nanomedicine, denoted as GO-PEG(TP), has been developed, comprising PEGylated nano graphene oxide co-loaded with photosensitizers (PS) and a two-photon compound, to combat cancer. The solutions of GO, GO-PEG, and GO-PEG(TP) (each with a concentration of 0.05 mg/mL of GO) were compared to demonstrate that this integrated therapeutic approach shows remarkable efficacy in targeting and eradicating 4T1 murine breast cancer cells, resulting in a significant induction of apoptosis among the cell population [220].

Gong et al. created nanocomposites by functionalizing magnetic graphene oxide (MGO) with triformyl cholic acid and folic acid (MGO-TCA-FA). This innovative approach establishes an efficient nanoplatform for photo-chemotherapy (PCT) for targeting liver cancer. The advantages of this platform include the capability for multiple-targeted drug delivery, drug release triggered by both pH levels and NIR, and remarkable efficiency in photothermal conversion [221]. GO is combined with PEG as a photothermal material to induce a heating effect in macrophages to enable its anti-tumor effect in vitro and in vivo [222].

A photothermal therapeutic agent has been developed using reduced graphene oxide for the targeted ablation of A549 lung cancer cells. The fabrication method involves a one-step, biocompatible process utilizing Memecylon edule leaf extract for the reduction of GO, ultimately yielding polyphenol-anchored reduced graphene oxide (RGO). RGO exhibits remarkable sensitivity to NIR irradiation, allowing precise in vitro targeting of lung cancer cells and delivering cytotoxic effects [223]. The plant extract of Salvia spinosa facilitates the conversion of graphene oxide (GO) to RGO. It significantly destroyed the PC cells (Panc02-H7) after exposing RGO-loaded PC cells to laser radiation [224]. A report evidenced the effects of a combination cancer therapy including RT (doses of 2 and 4 Gy) and PTT (808 nm laser irradiation) as a radio-photothermal therapy (RPTT) for a KB oral squamous carcinoma cell line in the presence of Fe_3_O_4_@Au/reduced graphene oxide (rGO) nanostructures (NSs) at different concentrations [225].

##### Photodynamic Therapy

PDT, a Food and Drug Administration (FDA)-approved cancer therapy, relies on photosensitizers that generate reactive oxygen species (ROSs) when exposed to specific light, effectively killing cancer cells. This minimally invasive treatment damages tumor vasculature, triggers an immune response, and boasts specificity and repeatability. To enhance its effectiveness, nanocarriers are utilized to deliver photosensitizers directly to the tumor site, making PDT a promising and targeted cancer treatment option.

GO was conjugated to amine-terminated six-armed PEG via amide formation and then loaded with Ce6 via supramolecular π-π stacking, which showed lower singlet oxygen generation efficiency than free Ce6. GO-PEG-Ce6 significantly enhances photodynamic cancer-cell-killing efficacy by facilitating an increased cellular uptake of Ce6 through the nanographene carrier [226]. Hyaluronic acid (HA)–GO conjugates, with a high loading of photosensitizers (Ce6), the PDT efficiencies of which were remarkably improved ∼10 times more than that of free Ce6, significantly influenced the co-treatment with an excess amount of HA polymers, illustrating their active targeting to HA receptors overexpressed on cancer cells [227]. A novel hybrid of GO and hypocrellin B (HB) generated efficient singlet oxygen via irradiation to damage tumor cells [228]. The interaction of methylene blue (MB) as a photosensitizer with GO has good performance in PDT during red-light-emitting diode (LED) irradiation, which showed cell-killing potential on MDA-MB-231 breast cancer cells [229]. Folic acid (FA) and chlorin e6 (Ce6) double-functionalized GO can penetrate rapidly into cancer cells and macrophages, exhibiting good photothermal properties and a high ROS generation capacity. Moreover, a combined effect of PTT and PDT leads to a higher killing efficiency toward different types of cells involved in cancer and other diseases [230].

Guo et al. developed a drug delivery system incorporating Paclitaxel (PTX) onto PEG-modified and oxidized sodium alginate (OSA)-functionalized GO nanosheets (NSs), called PTX@GO-PEG-OSA. The photothermal conversion ability was tested via subjecting GO-PEG-OSA NSs and PTX@GO-PEG-OSA NSs (GO-PEG-OSA concentration ~ 0.1 mg/mL) to 808 nm of NIR laser irradiation. After NIR irradiation, PTX@GO-PEG-OSA could generate excessive ROS, attack mitochondrial respiratory chain complex enzymes, reduce adenosine-triphosphate (ATP) supplements for P-glycoprotein(P-gp), and effectively inhibit P-gp’s efflux pump function, thereby inducing obvious antitumor effects on gastric cancer [231]. A pluronic-based graphene oxide-methylene blue (GO-MB/PF127) nanocomposite was activated by both 808 nm NIR light and a 660 nm LED source. In this system, the GO component induced photothermal ablation of cancer cells, while methylene blue (MB) generated singlet oxygen, thereby destroying cancer cells via oxidative stress in PDT [232].

Nitrogen-doped graphene oxide dots (NGODs) can effectively produce H_2_O_2_ under white light irradiation and their H_2_O_2_ rate is proportional to the ascorbic acid (AA) concentration. This AA-supplemented PDT effectively kills lung, head and neck, colon, and oral cancer cells; however, it is highly safe for normal cells [233]. Liu et al. reported that nanoscale graphene oxide (NGO) was synthesized and then loaded with gold nanoparticles (AuNPs) and thiol polyethylene glycol folic acid (SH-PEG1000-FA). Further modifications involved incorporating the photosensitizer MB or the anticancer drug 5-fluorouracil (5-Fu). These multifunctional nanoplatforms were designed to facilitate either photothermal–photodynamic synergy (NGO-AuNPs-FA/MB) or photothermal–chemotherapy synergy (NGO-AuNPs-FA/5-Fu). When exposed to laser irradiation, they demonstrated exceptional photodynamic and photothermal properties, resulting in excellent in vitro antitumor effects [234].

#### 3.3.3. Angiogenesis and Anti-Angiogenesis Therapy

In recent studies, GO showed angiogenesis or anti-angiogenesis properties. Mukherjee et al. demonstrated that the intracellular formation of reactive oxygen species and reactive nitrogen species as well as the activation of phospho-eNOS and phospho-Akt might be plausible mechanisms for GO and rGO-induced angiogenesis [235]. GO/polycaprolactone (PCL) nanoscaffolds are fabricated to evaluate their pro-angiogenic characteristics. The AKT-endothelial nitric oxide synthase (eNOS)-vascular endothelial growth factor (VEGF) signaling pathway might play a major role in the angiogenic process [236]. However, GO also affected the consumption of niacinamide, a precursor of energy carriers, and several amino acids involved in the regulation of angiogenesis. The combination of the physical hindrance of internalized GO aggregates, induction of oxidative stress, and alteration of some metabolic pathways leads to a significant antiangiogenic effect in primary human endothelial cells [237]. GO containing 6-gingerol (Ging) modified with chitosan (CS)-FA nanoparticles (Ging-GO-CS-FA) have pro-oxidant power against cancer cells by reducing the amount of the superoxide dismutase (SOD) gene, its average length, and the number of blood vessels on angiogenesis [238].

## 4. Frontiers in Pancreatic Cancer Organoids and Graphene Oxide Platform

### 4.1. Current Research of Graphene Oxide in 3D Culture Tumor

3D culturing represents a good model for restructuring tissue microenvironments. In recent years, more and more researchers have focused on nanoparticles and 3D culturing of spheroids or organoids. Herein, we summarize these studies, which evidence the varying effects of graphene-based nanoparticles on 3D culture cells mode (Table 2).

Graphene oxide selectively hinders the proliferation of cancer stem cells across a diverse range of tumor types. Utilizing the tumor-sphere assay, GO’s efficacy in inhibiting tumor-sphere formation was represented in six distinct cancer types, including breast, ovarian, prostate, lung, pancreatic, and glioblastoma cancers [239]. A microfluidic lab-on-a-chip system was effectively employed to establish four-day cultures of liver (HepG2), breast (MCF-7), and colon (HT-29) cancer cells in both 2D monolayer and 3D spheroid configurations. Notably, the removal of graphene oxide proved to be more straightforward from flat structures than from three-dimensional ones. The variations between 2D and 3D models may arise from distinct cellular responses contingent on their developmental context [240].

GO coated on the sidewalls of micro-wells fabricated from a cell-adhesion-resistive polymer was found to efficiently initiate the distinct donut-like formation of cancer cell spheroids. Vertically coated GO micropatterns (vGO-MPs) of varying sizes (100–250 µm) were observed on polymer platforms with HepG2 cells. The 150 µm-sized platform was found to efficiently and rapidly induce the formation of 3D spheroids in the absence of cell-adhesion proteins [241]. Grilli et al. analyzed the efficiency of 3D lung cancer spheroids combined with a minimally modified graphene oxide -based nanocarrier for siRNA delivery as a new system for cell transfection, which demonstrated the higher efficiency of spheroids compared to 2D models for transfection and the high potential of unmodified GO to carry siRNA [242].

GO flake interactions with both in vitro (U-87 MG three-dimensional spheroids, devoid of stromal or immune components) and in vivo (U-87 MG orthotopic xenograft) glioblastoma models were examined, which suggested that GO flakes can achieve a comprehensive and uniform distribution within glioblastoma tumors and serve as a potential approach to target their myeloid compartment. This opens possibilities for employing GO flakes as a platform to develop immunomodulation strategies against glioblastoma by specifically targeting macrophage and microglia compartments [243]. Graphene oxide inhibits the growth and malignancy of glioblastoma stem cell-like (GSC) spheroids through epigenetic mechanisms that drive the differentiation of GSCs, thereby reducing the malignancy of glioblastoma multiforme(GBM) [244]. GO loaded with PEG superparamagnetic iron oxide nanoparticles and grafted with methotrexate and stimuli-responsive linkers (GO-SPION-MTX) were developed for breast cancer treatment in 3D culture. These nanocomposites were internalized by cancer cells expressing folate receptors and demonstrated high cytotoxicity when subjected to NIR laser rays [245].

**Table 2 ijms-25-01066-t002:** The applications of graphene-based nanoparticles in 3D culture cells.

Type of Graphene-Based Nanoparticle	Categories of Origin	Function and Highlights	Ref.
Graphene oxide (GO)	liver (HepG2), breast (MCF-7) and colon (HT-29) cancer cells	Microfluidic Lab-on-a-Chip systems	[240]
GO	human brain	used multi-omics techniques to investigate the mechanisms of GO on lipid homeostasis in a 3D brain organoid model. Transcriptomics and lipidomics indicated that direct contact with GO altered lipid homeostasis through ER stress in 3D human brain organoids	[246]
GO	Inner ear organoids (IEOs)	promote cell–extracellular matrix interactions and cell–cell gap junctions, potential applications for drug testing	[247]
GO	U87, U251 GSCs and primary GSCs	GO could promote differentiation and reduce malignancy in GSCs via an unanticipated epigenetic mechanism	[244]
Graphene oxide (GO) flakes	Human Glioblastoma	GO flakes translocated deeply into the spheroids	[243]
Gold-graphene hybrid nanomaterial (Au@GO)	co-culture spheroids of HeLa/Ovarian cancer and HeLa/human umbilical vein endothelial cell (HUVEC)	Au@GO nanoparticles displayed selectivity towards the fast-dividing HeLa cells, which could not be observed to this extent in 2D cultures.	[248]
Graphene oxide (GO)-based nanocarrier for siRNA	lung cancer(A549)	high potential of unmodified GO to carry siRNA	[242]
Fluorescent chitosan/graphene oxide hybrid microspheres (GCS/GO)	human umbilical cord mesenchymal stem cells	The hybrid microspheres can support long-time stem cell expansion, autofluorescence also makes observing and tracking the stem cells’ behavior on the surface of microsphere scaffolds	[249]
HA-EDA-PHEA-DVS/GO composite nano gel	human colon cancer cells (HCT 116)	conduct thermal ablation of solid tumors	[250]
graphene oxide (GO) loaded with PEGylated superparamagnetic iron oxide nanoparticles and grafted with methotrexate and stimuli-responsive linkers (GO-SPION-MTX)	breast cancer cell	GO-SPION-MTX was internalized by the folate-receptor-positive cancer cells and induced high cytotoxicity on exposure to NIR laser rays	[245]
Herceptin-stabilized graphene	breast cancer cells (SKBR-3)	ultrasonic-assisted method in one-step synthesis, long-term stability in aqueous solutions	[251]
Graphene nanoplates	human endothelial cells, human brain perivascular pericytes, primary neurons, human astroglia cells, and primary microglia	spheroid bulk is formed by neural cells and microglia and the surface by endothelial cells and they upregulate key structural and functional proteins of the blood-brain barrier. These cellular constructs are utilized to preliminary screen the permeability of polymeric, metallic, and ceramic nanoparticles	[252]
Graphene quantum dots (GQDs)	human hepatoma cell line (HepG2 cells)	The chirality of GQDs (L/D-GQDs) was modification with L/D-cysteines. L-GQDs are more effective as nanocarriers for Doxorubicin delivery	[253]
Hydroxylated GQDs (OH-GQDs)	mice intestinal crypts	OH-GQD treatment significantly reduced the size of the surviving intestinal organoids.	[254]
reduced graphene oxide-branched polyethyleneimine-polyethylene glycol (rGO-BPEI-PEG)	uniformly sized neural stem cell (NSC)-derived neurospheres	Photothermal therapy (PTT) application of brain tumor spheroids generated by the microfluidic device using rGO-BPEI-PEG nanocomposite as the PTT agent	[255]
Magnetic nanoparticle-decorated reduced graphene oxide (m-rGO)	neuroblastoma cells (SH-SY5Y)	encapsulating SH-SY5Y to promote cell differentiation and induce oriented cell growth owing to its excellent biocompatibility and electrical conductivity	[256]
Reduced graphene oxide (rGO) was the carrier for the loading of doxorubicin (DOX) and chlorin e6 (Ce6) (rGO-PEG/Ce6 and rGO-PEG/DOX)	glioma cells (U87)	PTT showed great treatment efficacy in the 3D tumor spheroid mode than CT and PDT	[257]
Reduced graphene oxide-MXene (rGO-Mxene) hydrogel	epithelial adenocarcinoma, neuroblastoma, and fibroblasts	strong affinity of cellular protrusions (neurites, lamellipodia, and filopodia) to grow and connect along architectural network paths within the rGO-Mxene hydrogel, leading to control over macroscopic formations of cellular networks for technologically relevant bioengineering applications	[258]
Gelatin with methacryloyl groups (GeIMA) and reduced graphene oxide (rGO)	colon carcinoma cells (RKO)	GelMA with higher crosslink densities and promote proliferation	[259]
Vertically coated GO micropatterns (vGO-MPs)	human liver cancer cells (HepG2)	Cytophilic GO is selectively coated on the sidewalls of micro-wells fabricated from a cell-adhesion-resistive polymer to efficiently initiate distinct donut-like formation of cancer cell spheroids. Highly stable, the anticancer effects improved	[241]
Engineered carbon nanotubes (CNTs)	intestinal organoids	promoted the development of intestinal organoids over time, CNTs reduced the hardness of the extracellular matrix by decreasing the elasticity and increasing the viscosity	[260]
Nanowire (NW)-templated 3D fuzzy graphene (NT-3DFG)	primary E18 rat cortical tissues	a hybrid nanomaterial for remote, nongenetic, photothermal stimulation of 2D and 3D neural cellular systems.	[261]
3D interconnected graphene–carbon nanotube web (3D GCNT web)	glioma and healthy cortical cells	3D trajectories and velocity distribution of individual infiltrating glioma to be reconstructed with unprecedented precision	[262]
Three-dimensional graphene foam (3D-GFs)	neural stem cell (NSC)	3D-GFs can enhance the NSC differentiation towards astrocytes and especially neurons.	[263]
3D-SR-Bas with active biosensors: graphene field-effect transistor (GFET)	human cardiac spheroids (HUES9 hESCs)	provided continuous and stable multiplexed recordings of field potentials with high sensitivity and spatiotemporal resolution,	[264]

### 4.2. Current Research of Graphene Oxide in Pancreatic Cancer

In recent years, nanoparticles based on GO have been employed in pancreatic cancer research, demonstrating superior efficacy (Table 3). GO was employed as a gene delivery system for the simultaneous delivery of HDAC1 and K-Ras siRNAs, which target the HDAC1 gene and the G12C mutant K-Ras gene, respectively. This targeted delivery was specifically designed to affect pancreatic cancer cells, particularly MIA PaCa-2. Utilizing GO-based nanoformulations in conjunction with NIR light, tumor volume growth in mice was significantly reduced by up to 80%, which highlights the potential of GO-based nanocarriers in combining cancer gene therapy with photothermal effects for pancreatic cancer treatment [265].

Hybrid droplets containing gold–graphene oxide (Au-GO), doxorubicin, and zwitterionic chitosan (ZC) for the assembly of Au-GO@ZC-DOX stealth nanovesicles (NVs) were used to prevent macrophage opsonization, resulting in anti-cancer and anti-migration effects with high intracellular uptake in PANC-1 and MIA PaCa-2 cells [266]. The analysis of personalized biomolecular coronas (BC) with GO has demonstrated its potential for early cancer detection. In a study involving 50 subjects—half with pancreatic cancer and half being healthy volunteers—GO nanometric flakes were used to assess their BCs. The test achieved a remarkable 92% sensitivity in distinguishing cancer patients from healthy individuals, with an impressive area under the curve (AUC) of 0.96. These results highlight the promising role of GO-based BC analysis in early pancreatic cancer identification [267].

Sudhakara Prasad et al. developed affordable, disposable, paper-based immunosensors for the early quantitative detection of pancreatic cancer using the novel biomarker SGK269 (PEAK1). These immunosensors feature paper-based electrodes (PPEs) coated with GO, offering a specific diagnostic platform suitable for point-of-care and low-resource settings [268]. Graphene oxide nanosheets on a patterned gold surface were used to capture circulating tumor cells (CTCs) from blood samples of pancreatic cancer patients, as CTCs can be valuable biomarkers for disease diagnosis and progression [269].

Quagliarini et al. underscored the potential of leveraging the magnetic levitation of graphene oxide–protein complexes in conjunction with blood glucose levels for the early detection of PDAC. The observed significant variations in levitating nanosystems between controls and PDACs suggest the feasibility of employing this innovative approach as a multiplexed blood test for PDAC screening, especially in populations with hyperglycemia [270]. A positively charged lipid bilayer membrane was applied to reduced graphene oxide@gold nanostar (rGO@AuNS) for photoacoustic/photothermal dual-modal imaging-guided gene/photothermal synergistic therapy in pancreatic cancer. The combined photothermal and gene therapy, targeting the G12V mutant K-Ras gene, demonstrated remarkable anticancer and anti-liver metastasis efficacy in mice with pancreatic cancer tumors [271].

**Table 3 ijms-25-01066-t003:** The applications of GO-based nanoparticles in pancreatic cancer research.

Nanocarrier	Function	Model of Pancreatic Cancer	Type of Study	Reference
GO	selectively targets cancer stem cells (CSCs) of multiple cancer cell types	MIA PaCa-2	in vitro	[239]
GO	a multiplexed Maglev-based nanotechnology as a screening tool for PDAC in populations with hyperglycemia	plasma of patients	in vitro	[270]
GO	enhance the combined effect of hyperthermia and radiation treatment	-	in vitro	[272]
GO	investigate toxicity	BxPC-3, AsPC-1	in vitro	[273]
GO nanoflakes	cancer identification at early stages via analysis of the personalized biomolecular corona (BC)	plasma of patients	in vitro	[267]
GO nanosheets	nanoparticle-enabled blood test and serum levels of acute-phase protein detection	human blood	in vitro	[274]
GO sheets	synergistic analysis of protein corona and hemoglobin levels	plasma of patients	in vitro	[275]
GO-Au nanosheets	circulating tumor cells (CTCs) were captured with high sensitivity at a low concentration of target cells	blood of patient	in vitro	[269]
GO–Protein Corona Complexes	protein cornona detection, in vitro diagnostic (IVD) testing	plasma samples of PDAC patients	in vitro	[276]
GOQDs	facile pulsed-laser ablation in liquid (PLAL) technique for preparing GOQDs exhibited excellent optoelectronic properties	PANC-1	in vitro	[277]
GQD-HSA-Gem	drug delivery and bioimaging	PANC-1	in vitro	[278]
MWCNT-COOH/GO	use as two- and three-dimensional scaffolds to tissue engineer tumor models	PANC-1, BxPC-3, AsPC-1	in vitro	[279]
Nitrogen-doped graphene quantum dots (NGQDs)	pre-miR-132 detection for diagnosis	-	in vitro	[280]
PAH/FA/PEG/GO siRNA (HDAC1/K-Ras) complex	siRNA delivery, photothermal (808 nm), and gene therapy	MIA PaCa-2/Athymic nude mice (BALB/cASlac-nu)	in vitro, in vivo	[265]
Polymer-GO	Efficient capture and release of viable circulating tumor cells	PANC-1	in vitro	[281]
Reduced GO-gold-palladium (rGO-Au-Pd)	detect carbohydrate antigen 24-2 (CA242) marker	human serum	in vitro	[282]
rGO	Photothermal (980 nm)	mice Panc02-H7/C57BL/6 mice	in vitro, in vivo	[283]
RGO	S. spinosa leaf extract reduced the GO into RGO, photothermal (808 nm)	mice Panc02-H7	in vitro	[224]
RGO FET	identify early diagnostic biomarker miRNA10b	plasma samples	in vitro	[284]
rGO@AuNS-lipid (DODAB/DOPE-FA)	imaging: photoacoustic/photothermal; therapy: PTT/gene	Capan-1/Capan-1 tumor-bearing nude mice	in vitro, in vivo	[271]
ss-DNA@GoQdot@miR-141	GoQdot modified and thiolated single-stranded DNA detection probes (thiol-ss-DNA) on screen-printed electrodes (SPEs) interacted with the miR-141 marker, electrochemical biosensor	-	in vitro	[285]
Au-GO@ZC-DOX	high intracellular uptake, chemo-phototherapy	MIA PaCa-2, PANC-1	in vitro	[266]
Anti-PEAK1-GO-PPE	a low-cost electrochemical immunosensor on paper for the quantitative analysis of biomarker PEAK1	-	in vitro	[268]
AuNCs/GO	GO improved the sensitivity of AuNCs-based PEC immunosensors, Glypican-1 (GPC1), antigen (CEA), and glutathione (GSH) for early diagnosis	PANC-1/Balb/c-nu mice	in vitro, in vivo	[286]
carboxyl-GO	ultra-sensitive carboxyl-functionalized graphene oxide (GO-COOH)-based surface plasmon resonance (SPR) immunosensors using a carbohydrate antigen (CA) 199 (CA199) biomarker	blood of the patient	in vitro	[287]

### 4.3. Limitation of Current Organoids for Pancreatic Cancer Research

The current research indicates significant progress in utilizing pancreatic cancer organoids to simulate the pathophysiological processes of pancreatic cancer. However, several limitations remain. Firstly, prolonged culturing may alter organoid behavior, potentially compromising the ability to faithfully recapitulate tissue phenotypes and pathophysiological processes because of a potential loss of genetic diversity over extended periods [288,289]. Secondly, variations in the culture media’s composition can influence organoid behavior. While organoids can maintain complex cell interactions in cultures, changes in response to different culture conditions pose challenges in accurately interpreting experimental results and achieving study reproducibility [290]. Furthermore, capturing the full heterogeneity of the disease is very complex, especially given its intricate and diverse nature. While organoids derived from pancreatic cancer patients offer valuable insights, they may not fully encompass the entire spectrum of tumor heterogeneity. This limitation can be attributed to the selective growth of certain cell types in organoids and potential changes during the culturing process [67,291,292]. Lastly, existing pancreatic cancer organoid models often lack a fully functional vascular system, limiting their ability to faithfully replicate the tumor microenvironment. The absence of a vascular system hinders a comprehensive understanding of tumor growth, invasion, and treatment responses [94,103,293].

### 4.4. The Barriers of Graphene Oxide for Pancreatic Cancer Research

GO also has several limitations and challenges in pancreatic cancer research that necessitate further investigation. Firstly, there is a critical need for comprehensive, long-term biosafety studies to assess the potential impact on human health. The toxicity and potential side effects of GO on healthy cells remain a concern and require thorough examination [294]. The mechanism of GO degradation and excretion is not well elucidated, and understanding these processes is essential for assessing its long-term safety, particularly as the oxide part of GO may induce the generation of reactive oxygen species (ROS), influencing cytotoxicity [295]. The sensitivity of GO’s conductivity and capacity to respond to local electrical and chemical perturbations is crucial for its effectiveness in cancer research. Although GO can absorb light in the near-infrared (NIR) region, its absorption is not sufficiently strong, raising questions about its efficacy in photothermal applications for pancreatic cancer treatment. Immunogenicity is a vital consideration [296,297]. Additionally, the water solubility of GO needs improvement, and future studies should focus on enhancing nanocarrier biocompatibility and stability, reducing size-related toxicity, and preventing agglomeration during biomedical applications [298].

GO composites, known for their durability, may persist and accumulate in environments such as water bodies and soil, potentially leading to ecological imbalances due to their limited degradability [299]. Studies indicate that these nanoparticles can be toxic to aquatic organisms, causing oxidative stress and cellular damage, and may also lead to bioaccumulation in the food chain, impacting the transport and toxicity of other environmental contaminants [300].

The preparation of GO composites faces significant challenges in achieving reproducibility and uniformity because of factors such as the degree of oxidation, defects, and variations in the size and thickness of GO sheets [301,302]. Scaling up production and integrating materials homogenously, while maintaining GO’s inherent properties and ensuring cost-effectiveness, are major hurdles for their widespread commercialization [301].

In conclusion, while GO holds promise in pancreatic cancer research, these unresolved issues and challenges underscore the importance of continued investigation and optimization for its safe and effective application in biomedical settings.

### 4.5. Future of Pancreatic Cancer Organoids and Graphene Oxide Model Systems

In recent years, there has been significant progress in personalized therapy for pancreatic cancer, primarily driven by advancements in organoid technology. This approach has become increasingly important, especially considering that the majority of pancreatic malignancies are pancreatic ductal adenocarcinomas (PDAC). The groundwork for this technology was laid by Clevers’ lab, which has been pivotal in the development of pancreatic tissue organoids. The pioneering work in this field was published in 2013 by Huch et al., who successfully developed pancreatic organoids from isolated pancreatic ducts in mice [303]. This breakthrough provided a foundational model for studying pancreatic diseases. Following this, in 2015, Boj et al. made a significant advancement by reporting the first tumor organoids derived from human PDAC, marking a substantial step forward in personalized cancer treatment [24].

As we discussed the current limitations of pancreatic cancer organoids in the previous section, an advanced organoids model must necessarily be improved in the future. PDO can mimic various disease phases in PDAC; however, genetic diversity could be lost over extended periods. 3D printing allows for the creation of organoids with intricate structures that closely mimic the architecture [304]. In addition, it is produced in a standardized, reproducible manner, which is important for consistent research outcomes. Scalability is also enhanced, allowing for specific scale production of pancreatic cancer organoids. The realistic structure and function of 3D-printed organoids make them ideal for more accurate drug testing and disease modeling [305,306].

GO, as nanocarriers, have delivery capabilities which enable precise dosing and the targeted delivery of therapeutics within the organoid model. This can lead to more accurate assessments of drug efficacy and toxicity for individual patients. The integration of drug screening with CRISPR-Cas9 genome editing, as explored by Hirt et al., represents a significant advance in studying drug–gene interactions in PDAC [307]. This approach enables the modification of genes associated with disease and drug resistance and the introduction of tumor-suppressing genes, showing great promise in pancreatic cancer treatment. Crucial to this method is the development of safe, non-viral vectors for efficient nucleic acid delivery, a focus of intense research interest [308]. GO has emerged as a promising candidate in this field. Its ability to load both single-stranded DNA and RNA, despite its overall negative charge, is facilitated by hydrophobic and π-π stacking interactions between nucleic acid nucleobases and GO’s hexagonal carbon lattice [309]. This interaction may be enhanced by partial deformation of the nucleic acid’s double helix, improving adsorption onto GO’s surface.

Moreover, environmental conditions such as high salt concentrations and low pH have been shown to significantly augment the binding of double-stranded nucleic acids onto GO, suggesting its potential as an effective vector in gene therapy for pancreatic cancer [310]. Natural killer (NK) cells, when combined with antibodies that induce antibody-dependent cell-mediated cytotoxicity (ADCC), present a highly promising therapeutic approach for pancreatic cancer. Beelen et al. demonstrated that the ADCC-inducing antibodies avelumab (anti-PD-L1) and trastuzumab (anti-HER2) enhance NK cell-mediated cytotoxicity, leading to increased death of organoid cells [311]. Furthermore, when combined with GO, it is possible to create a more complex and interactive model that includes not just cancer cells, but also the surrounding stroma and immune cells.

Research has investigated the photothermal effects of GO under near-infrared irradiation. GO can convert light energy into heat energy, which can be used to increase the temperature of tumor tissue, effectively killing tumor cells [312]. The extracellular matrix (ECM) plays a crucial role in this process, acting as a barrier within the tumor microenvironment. In 3D culture organoids, which mimic this structure, the penetration of GO and its ability to transport heat are essential. This heat can damage the DNA of cancer cells, making GO’s properties vital for effective photothermal therapy [313,314,315]. Accordingly, the photodynamic therapy of GO also can be explored in organoids for personalized pancreatic cancer treatment.

In addition, malignant pancreatic cancer is often subject to late diagnosis. The unique optical properties of GO, such as fluorescence quenching, can be utilized in the development of optical biosensors for pancreatic cancer detection [316]. These sensors can offer rapid and non-invasive diagnosis, which is critical for early detection of this aggressive cancer. Another study focused on the protein corona (PC) formed around GO nanoflakes in human plasma. This research has implications for early cancer detection, where changes in the concentration of typical biomarkers are often too low to be detected by blood tests. The study suggests that the personalization of PC in GO can be maximized, enhancing the profiling between cancer vs. non-cancer patients [276].

Besides these properties mentioned above, the PH sensitivity, angiogenesis, and anti-angiogenesis ability of GO also are necessary for pancreatic cancer research. For example, the microenvironment of cancer is always more acidic than other normal parts [317,318]. The vessel formation is important for the tumor to obtain nutrition. Integrating GO with PDO can potentially enhance the effectiveness of drug delivery systems, improve imaging and monitoring capabilities, and offer new therapeutic strategies such as photothermal therapy. As artificial intelligence (AI) has been developing very quickly recently, more and more complex analyses and restructure models will help to improve the function of organoids and GO. The combination of these two platforms could lead to more accurate and personalized treatments for pancreatic cancer, addressing the need for advanced and effective therapeutic approaches in PDAC.

## 5. Conclusions

In conclusion, this review underscored the promising potential of leveraging the synergy between organoids and GO in advancing pancreatic cancer treatment. While organoids offer a faithful replication of the cancer’s genetic diversity and microenvironment, GO provides versatile applications in targeted drug delivery and diagnostics. Despite their individual merits, both organoids and GO face limitations, such as the complexity of replicating tumor heterogeneity and the biosafety concerns of GO. Looking forward, the integration of these two platforms could revolutionize the study of pancreatic cancer, aiding in personalized medicine, effective drug screening, and biomarker discovery. However, further research is necessary to address current challenges and optimize this combined approach for clinical application, potentially offering new hope in a field marked by limited treatment options and poor prognosis.

## Figures and Tables

**Figure 1 ijms-25-01066-f001:**
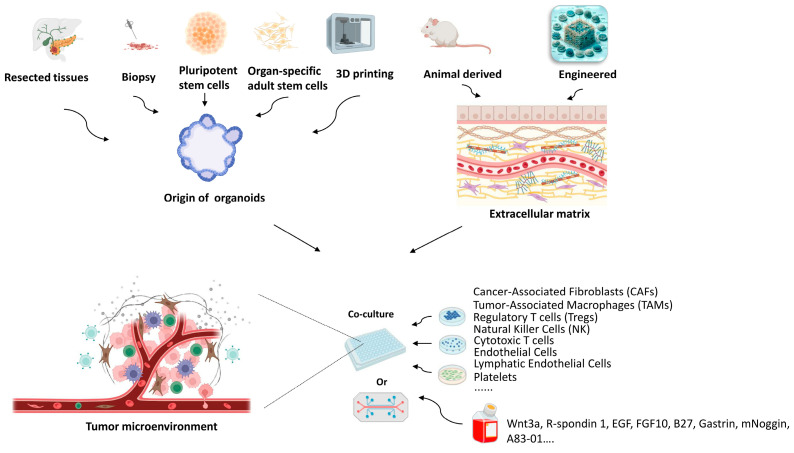
Comprehensive overview of organoid development and tumor microenvironment modeling. The diverse origins of organoids, including resected tissues, biopsies, pluripotent stem cells, organ-specific adult stem cells, and 3D printing techniques are highlighted. The composition of the extracellular matrix (ECM) distinguishes between animal-derived and engineered components. In the context of the tumor microenvironment, the figure showcases the integration of organoids with elements such as culture medium, non-neoplastic cells, and angiogenesis, emphasizing the utility of organoids in replicating complex biological interactions. (Epidermal growth factor, EGF; Fibroblast growth factor 10, FGF10; Mouse Noggin Recombinant Protein, mNoggin; A83-01, Selective inhibitor of TGF-βRI, ALK4, and ALK7).

**Figure 2 ijms-25-01066-f002:**
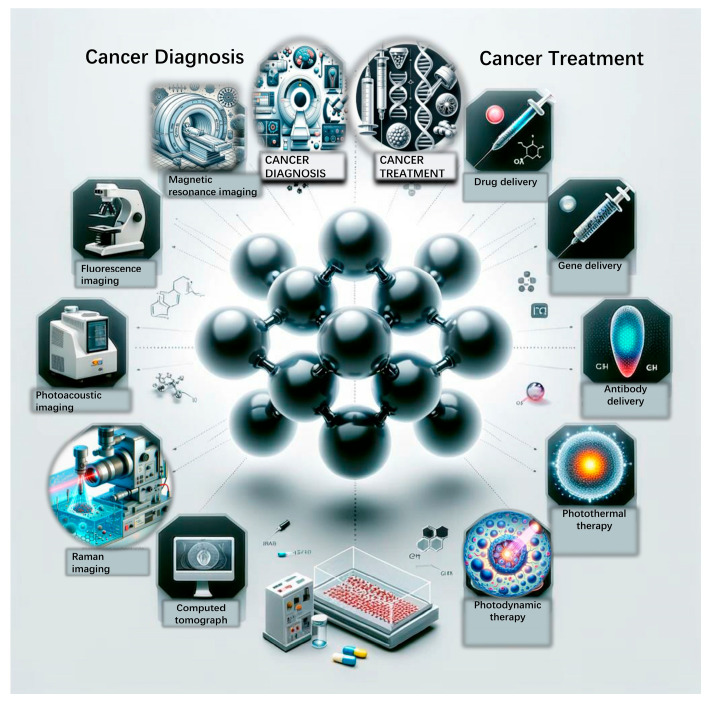
Schematic overview of graphene oxide applications in cancer diagnosis and treatment. Graphene oxide is widely utilized for its unique physical and chemical properties. Surrounding this central structure of GO are various methodologies employed in cancer diagnostics and therapeutics. On the left, diagnostic techniques are showcased: magnetic resonance imaging (MRI), fluorescence imaging, photoacoustic imaging, Raman imaging, and computed tomography (CT), highlighting the multifaceted applications of graphene oxide in enhancing imaging modalities. On the right, treatment approaches are depicted: drug delivery, gene delivery, antibody delivery, photothermal therapy, and photodynamic therapy, indicating the versatility of graphene oxide as a carrier and its role in targeted treatment strategies.

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
