# Peer review of "Graphene Oxide Nanoparticles and Organoids: A Prospective Advanced Model for Pancreatic Cancer Research"

_ijms, 2024, doi:10.3390/ijms25021066_

Round 1

Reviewer 1 Report

Comments and Suggestions for Authors

The current review article is an interesting manuscript on graphene oxide nanoparticles and organoids for pancreatic cancer treatment and diagnosis. Many relevant studies were included, and the review is overall complete in what concerns addressed topics. Hence, I only advise for a few changes before acceptance for publication:

- The abstract should be improved, by providing more relevant information on the general conclusions of the review;

- In the introduction section, more should be said about current therapies for pancreatic cancer and their limitations, hence better explaining the need for new improved therapies, such as nanoparticles;

- Although in Figure 1 image quality is good, it is small, making it hard to read and interpret, and hence should be enlarged;

- On table 2, in the column “Type of Study”, some cells say “in vivo, in vivo”, why the repetition?;

- Images composed of the cited article’s original results should be added (with the appropriate permissions from the original journals), for better reader visualization:

- Citations should be better placed, since some sentences, and even some paragraphs, have none.

Author Response

Dear Editor,

            We would like to thank the Reviewers for their thoughtful comments and efforts towards improving our manuscript. We appreciate their time invested in improving our work. We did our best to change manuscript according to suggestions. Please find below point-to-point responses to the Reviewers’ points.

Reviewer1: The current review article is an interesting manuscript on graphene oxide nanoparticles and organoids for pancreatic cancer treatment and diagnosis. Many relevant studies were included, and the review is overall complete in what concerns addressed topics. Hence, I only advise for a few changes before acceptance for publication:

- The abstract should be improved, by providing more relevant information on the general conclusions of the review;

Authors response: Thank you very much for this suggestion. We revised our abstract as indicated in the revised manuscript.

- In the introduction section, more should be said about current therapies for pancreatic cancer and their limitations, hence better explaining the need for new improved therapies, such as nanoparticles;

Authors response: Thank you very much for this advice. We added current treatments for pancreatic cancer and their disadvantages, which marked these sentences in introduction.

‘’Currently, standard treatment modalities for PDAC include surgery, chemotherapy, and radiation therapy. While surgical resection offers the best chance for a cure, only a small fraction of patients is eligible for this option. The majority of pancreatic cancer cases are diagnosed at an advanced stage, rendering surgery infeasible[4]. The mainstay of treatment for advanced pancreatic cancer, chemotherapy, has limited efficacy. Drugs like gemcitabine have been the standard, but they offer only modest improvements in survival and are frequently associated with significant side effects. In addition, as it is a broadly used drug, almost all pancreatic cancer patients eventually develop resistance to this drug[5]. FOLFIRINOX (oxaliplatin, irinotecan, leucovorin, 5-fluorouracil) could directly improve overall survival(OS) rate of patients with metastatic pancreatic tumor (HR 0.76, 95% Cl 0.67–0.86, p<0.001) but had no benefit on progression-free survival(PFS)[6]. Standard radiotherapy options, which typically deliver 40 to 60 Gy in 1.8–2.0 Gy fractions, offer minimal to no survival benefit for patients with locally advanced unresectable pancreatic cancer (LAPC) who have undergone chemotherapy[7]. Its role is limited due to the proximity of the pancreas to critical organs, which increases the risk of damage to surrounding tissues. Therefore, these limitations of treatment underscore the urgent need for innovative therapeutic strategies.’’

- Although in Figure 1 image quality is good, it is small, making it hard to read and interpret, and hence should be enlarged;

Authors response: Thank you very much for this suggestion. We have enlarged the text on the Figure 1.

- On table 2, in the column “Type of Study”, some cells say “in vivo, in vivo”, why the repetition?;

Authors response: Thank you for noticing that. We have checked and corrected these mistakes in the marked manuscript.

- Images composed of the cited article’s original results should be added (with the appropriate permissions from the original journals), for better reader visualization:

Authors response: Thank you for pointing this out. We have revised and two Figures in manuscript are original.

- Citations should be better placed, since some sentences, and even some paragraphs, have none.

Authors response: Thank you for pointing this out. In these sections, we reference several relevant publications as follows, which are also marked in the manuscript:

GO is characterized by its chemical and mechanical stability, biocompatibility, and a two-dimensional structure that allows for extensive surface modification[11]. These modifications can be tailored with various functional groups like epoxide, hydroxyl, and carboxyl, enabling the attachment of biomolecules such as proteins, DNA, and RNA[12]. This adaptability makes GO an attractive candidate for a range of applications, including drug and gene delivery, phototherapy, and bioimaging. In medical research, GO's ability to be used in drug delivery systems, diagnostics, tissue engineering, and gene transfection is particularly noteworthy[13]. Its solubility in water and intrinsic fluorescence properties in the visible/near-infrared spectrum enhance its suitability for these applications[14,15].

Firstly, prolonged culture may alter organoid behavior, potentially compromising their ability to faithfully recapitulate tissue phenotypes and pathophysiological processes due to a potential loss of genetic diversity over extended periods[295,296]. Secondly, variations in culture media composition can influence organoid behavior. While organoids can maintain complex cell interactions in culture, changes in response to different culture conditions pose challenges in accurately interpreting experimental results and achieving study reproducibility[297]. Furthermore, it is indeed a complex challenge to capture the full heterogeneity of the disease, especially given its intricate and diverse nature. While organoids derived from pancreatic cancer patients offer valuable insights, they may not fully encompass the entire spectrum of tumor heterogeneity. This limitation can be attributed to the selective growth of certain cell types in organoids and potential changes during the culturing process.[298-300]. Lastly, current pancreatic cancer organoid models often lack a fully functional vascular system, limiting their ability to faithfully replicate the tumor microenvironment. The absence of a vascular system hinders the comprehensive understanding of tumor growth, invasion, and treatment responses[94,103,301].

The mechanism of GO degradation and excretion is not well elucidated, and understanding these processes is essential for assessing its long-term safety, particularly as the oxide part of GO may induce the generation of reactive oxygen species (ROS), influencing cytotoxicity[303]. The sensitivity of GO's conductivity and capacity to respond to local electrical and chemical perturbations is crucial for its effectiveness in cancer research. Although GO can adsorb light in the near-infrared (NIR) region, its absorption is not sufficiently strong, raising questions about its efficacy in photothermal applications for pancreatic cancer treatment. Immunogenicity is vital considerations[304,305]. Additionally, the water solubility of GO needs improvement, and efforts should focus on enhancing nanocarrier biocompatibility and stability, reducing size-related toxicity, and preventing agglomeration during biomedical applications[306].

3D printing allows for the creation of organoids with intricate structures that closely mimic the architecture[313]. In addition, it produced in a standardized, reproducible manner, which is important for consistent research outcomes. Scalability is also enhanced, allowing for specific scale production of pancreatic cancer organoids. The realistic structure and function of 3D printed organoids make them ideal for more accurate drug testing and disease modeling[314,315].

This heat can damage the DNA of cancer cells, making GO's properties vital for effective photothermal therapy[322-324]. Accordingly, photodynamic therapy of GO also can be explored in organoids for personalized pancreatic cancer treatment.

In addition, malignant pancreatic cancer is because of late diagnosis. The unique optical properties of GO, such as fluorescence quenching, can be utilized in the development of optical biosensors for pancreatic cancer detection[325].

Except these properties mentioned above, PH-sensitivity, angiogenesis and anti-angiogenesis ability of GO also are necessary for pancreatic cancer research. For example, the microenvironment of cancer is always acid than other normal parts[326,327].

Reviewer 2 Report

Comments and Suggestions for Authors

Dear Authors,

An interesting review, congratulations!

I have several comments, which I am sure will make the main goal of your review paper more clear.

1. Please add a sentence or two to the abstract, explaining how exactly is GO coming into contact with the tumor? How is it delivered? Is the solution of GO injected in vivo, or only in vitro? Or is it deposited on a model cell culture (organoid?) by spraying, or added into the solution? Please also explain in the text in greater detail, how it happens.

2. Which concentrations of GO are used? Please mention in the text.

3. Please mention if flakes are used, or a monolayer (how big)?

4. Flake sizes? The miscibility of flake solution depending on the flake sizes? May it happen, that the concentrations are high and GO flakes stick together? It will affect the way GO adsorbs the tissues.

I recommend you raise these questions in the text, and if not studied yet, mention it clearly. Open questions may be picked by interested scientists and your paper will reach wider audience and be more extensively cited.

Overall I see your paper interesting and useful for a wider audience.

I suggest accepting paper after incorporation of the abovementioned questions.

Sincerely,

Reviewer

Author Response

Dear Editor,

            We would like to thank the Reviewers for their thoughtful comments and efforts towards improving our manuscript. We appreciate their time invested in improving our work. We did our best to change manuscript according to suggestions. Please find below point-to-point responses to the Reviewers’ points.

Reviewer 2: Dear Authors,

An interesting review, congratulations!

I have several comments, which I am sure will make the main goal of your review paper more clear.

  1. Please add a sentence or two to the abstract, explaining how exactly is GO coming into contact with the tumor? How is it delivered? Is the solution of GO injected in vivo, or only in vitro? Or is it deposited on a model cell culture (organoid?) by spraying, or added into the solution? Please also explain in the text in greater detail, how it happens.

Authors response: Thank you very much for your interesting questions. We revised our abstract (as indicated in the revised manuscript) and explain a bit detailed information for GO delivered, which marked in manuscript (Introduction).

‘’In the context of tumor therapy, GO is often used as a nano-carrier due to its ability to load a large number of hydrophobic drugs containing benzene rings. Its nano-network structure and hydrophobicity play a crucial role in this process[16,17]. Additionally, GO's surface properties can be affected by pH changes[18]. It remains stable at a neutral pH but becomes less stable at more extreme pH values[19]. This characteristic is particularly useful in cancer therapy, as tumor tissues generally have a more acidic microenvironment compared to normal tissues. Therefore, at lower pH values, such as those found in tumor tissues, the protonation weakens the hydrogen bond interaction between the drug and GO. This pH-sensitive property can be incorporated into the design of an anticancer drug delivery system, making GO an intelligent nano-carrier that releases drugs more effectively in the acidic environment of tumor tissues[20]. In addition, it is often functionalized with various components, such as synthetic polymers like polyethylene glycol (PEG) or natural polymers[21]. This functionalization can be achieved through covalent modification, which may alter the original structure of GO, or non-covalent methods, which do not affect its native structure. These modifications enhance the stability of GO in physiological solutions and facilitate its use in drug delivery[22].

The specific method of delivery (injection, deposition on cell cultures, etc.) would depend on the particular study design and the targeted application within the realm of cancer therapy.

  1. Which concentrations of GO are used? Please mention in the text.

Authors response: Thank you very much for your questions. We added concentration of GO that used in publications. And marked them in manuscript.

  1. Please mention if flakes are used, or a monolayer (how big)?

Authors response: Thank you very much for your suggestion. We checked and revised them as marked in the manuscript.

Here, dendrimers featuring amino group caps (DEN) are skillfully grafted onto GO nanosheets which lateral sizes in the range of 40–380 nm (mean size ∼ 175 nm)

Afua A. Antwi-Boasiako and colleagues reported the use of bioconjugated 2D graphene oxide (bio-GO) nanostructures, with the average lateral size of the layered GO sheets being approximately 0.08μm – 0.1μm.

GO nanosheets with average diameter of ~100 nm and a thickness of ~1.2 nm were prepared via a modified Hummer’s method

A de novo drug delivery nanosystem(~128nm) based on gold nanoparticles (GNPs), decorated PEG, and folate (FA)-conjugated GO was designed to load with doxorubicin hydrochloride (DOX) as a model anticancer drug

The CS/Dex functionalized GO nanocomposites (GO-CS/Dex) exhibited a diameter of about 300 nm and a thickness of 60 nm, which had a strong cytotoxicity to the cancer cells[198].

  1. Flake sizes? The miscibility of flake solution depending on the flake sizes? May it happen, that the concentrations are high and GO flakes stick together? It will affect the way GO adsorbs the tissues.

Authors response: Thank you very much for your interesting questions. The miscibility of GO in solutions can be influenced by flake size. Smaller flakes, especially monolayers, tend to form more homogeneous dispersions. This is due to the reduction in van der Waals forces that lead to aggregation in larger, multi-layered flakes. At higher concentrations, there is a possibility that GO flakes may aggregate. This aggregation can be influenced by various factors including flake size, sonication time, and the specific conditions of the solution. The tendency of GO to aggregate can affect its interaction with tissues, especially in biomedical applications.

Research has shown that the size of GO flakes can affect how different cell types interact with them. For instance, in liver cells, GO with larger lateral sizes was found to have stronger cytotoxic effects compared to smaller-sized GO. This size dependency is crucial because it impacts the internalization mechanism of GO in cells and the subsequent biological responses, such as the induction of cell death pathways.

The aggregation of GO in biological environments can alter its biological properties, affecting its internalization in cells. Studies have indicated that when GO aggregates, its efficiency to enter cells can be altered. For example, in lung cells, larger GO particles have shown a higher affinity towards cells, and samples incubated with serum (which may cause aggregation) entered the cells with lesser efficiency compared to non-incubated counterparts.

Reference:

Gacka, E., Majchrzycki, Ł., Marciniak, B. et al. Effect of graphene oxide flakes size and number of layers on photocatalytic hydrogen production. Sci Rep 11, 15969 (2021). https://doi.org/10.1038/s41598-021-95464-y

Li J, Wang X, Mei KC, Chang CH, Jiang J, Liu X, Liu Q, Guiney LM, Hersam MC, Liao YP, Meng H, Xia T. Lateral size of graphene oxide determines differential cellular uptake and cell death pathways in Kupffer cells, LSECs, and hepatocytes. Nano Today. 2021 Apr;37:101061. doi: 10.1016/j.nantod.2020.101061. Epub 2020 Dec 24. PMID: 34055032; PMCID: PMC8153408.

Dabrowski B, Zuchowska A, Kasprzak A, Zukowska GZ, Brzozka Z. Cellular uptake of biotransformed graphene oxide into lung cells. Chem Biol Interact. 2023 May 1;376:110444. doi: 10.1016/j.cbi.2023.110444. Epub 2023 Mar 10. PMID: 36906140.

I recommend you raise these questions in the text, and if not studied yet, mention it clearly. Open questions may be picked by interested scientists and your paper will reach wider audience and be more extensively cited.

Overall I see your paper interesting and useful for a wider audience.

I suggest accepting paper after incorporation of the abovementioned questions.

Authors response: We thank you the Reviewer for kind comments on our manuscript.

Sincerely Yours,

Authors

Reviewer 3 Report

Comments and Suggestions for Authors

I think this paper is a little confused. It describes both an interesting preclinical testing method using Organoids and the specific testing for clinical applications of Graphene oxide and related composites. The paper also has detailed sections for more general applications of GO with prostate cancer diagnostics and treatment. Perhaps the title should be changed to better display the general nature of this review.

In section 3.1 add a section in the synthesis of GO and r-GO and how it compares to graphene, graphene quantum dots and a general description of the applications of these materials in areas like nanosensors, environmental applications and composite materials.

You should separate out section 3.1.8 from 3.1 and put it into a biomedical applications section.

Particular emphasis should be made in section 3.2.1 on the importance of magnetic nanoparticle size used in the GO-MNP composites for example in reference 128 the particles are superparamagnetic while in reference 129 they are soft ferrimagnetic with possible effects on their behavior in the laboratory and the body.

Section 3.2 is stated to be about cancer diagnosis but you also have a lot of indications of the use of the same materials in treatments. Can you please make sure to separate out the different sections.

Sections 4.3 and 4.4 have no references. Please include some references to work in nano toxicology and potential damage to the environment as well as documented problems with the preparation of GO composites for these application.

Section 4.5 should be shifted to a discussion section.

Please revise the use of correct scientific notation. Make sure to use subscripts for chemical formulae. 

Author Response

Dear Editor,

            We would like to thank the Reviewers for their thoughtful comments and efforts towards improving our manuscript. We appreciate their time invested in improving our work. We did our best to change manuscript according to suggestions. Please find below point-to-point responses to the Reviewers’ points.

Reviewer 3:

I think this paper is a little confused. It describes both an interesting preclinical testing method using Organoids and the specific testing for clinical applications of Graphene oxide and related composites. The paper also has detailed sections for more general applications of GO with prostate cancer diagnostics and treatment. Perhaps the title should be changed to better display the general nature of this review.

Authors response: Thank you very much for your comment. In our manuscript, we have explored the applications of organoids in cancer research, focusing on their fundamental characteristics and current models. Additionally, we have detailed the role of GO and its applications in cancer therapy. We have comprehensively summarized existing research on the integration of GO with 3D culture systems, particularly emphasizing its use in pancreatic cancer research. Furthermore, we propose that combining GO with 3D cultured organoids could be an effective model for pancreatic cancer studies. This manuscript also thoroughly analyzes the advantages and limitations of these approaches, offering a balanced view of their potential in advancing pancreatic cancer research. We improved our title “Graphene Oxide Nanoparticles and Organoids: A Prospective Advanced Model for Pancreatic Cancer Research".

Reviewer 3:

In section 3.1 add a section in the synthesis of GO and r-GO and how it compares to graphene, graphene quantum dots and a general description of the applications of these materials in areas like nanosensors, environmental applications and composite materials.

Authors response: Thank you for your valuable suggestion. We have included information about graphene, GO, r-GO, and GQDs in Section 3.1 to more effectively explain their relationship with the properties of GO.

Graphene, a two-dimensional honeycomb lattice of carbon atoms, is typically synthesized via chemical vapor deposition or mechanical exfoliation of graphite, noted for its exceptional electrical, mechanical, and thermal properties[113]. GO created using the Hummers method, introduces oxygen-containing functional groups to graphite, resulting in a material with a large surface area and functional versatility, albeit with reduced electrical conductivity compared to graphene[114,115]. Reduced Graphene Oxide (r-GO), obtained by removing these oxygen groups from GO, restores some of graphene's electrical and structural features[116]. Graphene Quantum Dots (GQDs), small graphene fragments synthesized through top-down or bottom-up methods, exhibit unique optical and electronic properties due to quantum confinement and edge effects[117]. Both GO and r-GO, rich in functional groups, are adaptable for diverse modifications and applications, particularly in biosensors and environmental remediation, while GQDs find use in fluorescence-based sensors and electronics[118]. Graphene's integration into composites boosts their mechanical, thermal, and electrical characteristics, with r-GO also being leveraged in similar applications for its balance between conductivity and functional compatibility[119]. We discussed the detailed properties of GO in the following sections.

Reviewer 3:

You should separate out section 3.1.8 from 3.1 and put it into a biomedical applications section.

Authors response: Thank you very much for your suggestion. We revised them and put it in treatment part as marked in the manuscript.

‘’3.3.3. Angiogenesis and anti-angiogenesis therapy

In recent studies, GO showed angiogenesis or anti-angiogenesis properties. Mukherjee et al., demonstrated that the intracellular formation of reactive oxygen species and reactive nitrogen species as well as activation of phospho-eNOS and phospho-Akt might be the plausible mechanisms for GO and rGO induced angiogenesis[236]. GO/polycaprolactone (PCL) nanoscaffolds are fabricated to evaluate the pro-angiogenic characteristic. The AKT-endothelial nitric oxide synthase (eNOS)-vascular endothelial growth factor (VEGF) signaling pathway might play a major role in the angiogenic process[237]. However, GO also affected the consumption of niacinamide, a precursor of energy carriers, and several amino acids involved in the regulation of angiogenesis. The combination of the physical hindrance of internalized GO aggregates, induction of oxidative stress, and alteration of some metabolic pathways leads to a significant antiangiogenic effect in primary human endothelial cells[238]. GO containing 6-gingerol (Ging) modified with chitosan (CS)-FA nanoparticles (Ging-GO-CS-FA) that have pro-oxidant power against cancer cells by reducing the amount of the superoxide dismutase (SOD) gene, the average length, the number of blood vessels on angiogenesis[239].’’

Reviewer 3:

 Particular emphasis should be made in section 3.2.1 on the importance of magnetic nanoparticle size used in the GO-MNP composites for example in reference 128 the particles are superparamagnetic while in reference 129 they are soft ferrimagnetic with possible effects on their behavior in the laboratory and the body.

Authors response: Thank you very much for your detailed suggestion. We added the influence of magnetic nanoparticle in GO-MNP composites.

‘’The size of magnetic nanoparticles (MNPs) in graphene oxide (GO) composites critically influences their performance, with smaller MNPs enhancing surface reactivity, ensuring superparamagnetic behavior, and improving biological penetration and distribution, while also affecting stability and toxicity profiles[151]. This size-dependent variation in physical and magnetic properties is fundamental in tailoring GO-MNP conjugates for specific applications, particularly in biomedical fields like MRI[152].’’

Reviewer 3:

Section 3.2 is stated to be about cancer diagnosis but you also have a lot of indications of the use of the same materials in treatments. Can you please make sure to separate out the different sections.

Authors response: Thank you for your insightful suggestion. It is indeed true that recent research on GO conjugates predominantly focuses on multifunctional applications, often integrating diagnostic and therapeutic capabilities. This trend makes it challenging to isolate instances of nanoparticles used solely for diagnostic purposes, as such examples are more prevalent in older studies. While our aim is to delineate these functions as clearly as possible, the evolving nature of GO research, which leans towards combinatorial functionalities, should be taken into account to reflect the current and forward-looking scope of the field in our review.

Reviewer 3:

 Sections 4.3 and 4.4 have no references. Please include some references to work in nano toxicology and potential damage to the environment as well as documented problems with the preparation of GO composites for these application.

Authors response: Thank you for your good suggestion. We have incorporated this content into the manuscript as indicated. Additionally, we have cited relevant publications in Sections 4.3 and 4.4, which are marked in manuscript.

‘’GO composites, known for their durability, may persist and accumulate in environments like water bodies and soil, potentially leading to ecological imbalances due to their limited degradability[307]. Studies indicate that these nanoparticles can be toxic to aquatic organisms, causing oxidative stress and cellular damage, and may also lead to bioaccumulation in the food chain, impacting the transport and toxicity of other environmental contaminants[308].

The preparation of GO composites faces significant challenges in achieving reproducibility and uniformity, due to factors such as the degree of oxidation, defects, and variations in size and thickness of GO sheets[309,310]. Scaling up production and integrating materials homogenously, while maintaining GO's inherent properties and ensuring cost-effectiveness, are major hurdles for their widespread commercialization[309].’’

Reviewer 3:

Section 4.5 should be shifted to a discussion section.

Authors response: Thank you for your insightful suggestion regarding the structure of our review. We appreciate your recommendation to move "Section 4.5. Future of Pancreatic Cancer Organoids and Graphene Oxide Model System" to a separate discussion section. However, we believe that this section serves as an integral part of "Section 4. Frontiers in Pancreatic Cancer Organoids and Graphene Oxide Platform," where it contextually aligns with the ongoing narrative. Isolating it into a separate discussion could disrupt the flow and coherence of the content, as it directly relates to the advancements discussed in Section 4. We have ensured that the content within Section 4.5 is seamlessly integrated and relevant to the overarching theme of the section, thus providing a comprehensive and focused discourse on the subject.

Reviewer 3:

Please revise the use of correct scientific notation. Make sure to use subscripts for chemical formulae.

Authors response: Thank you very much for your kindly reminder. We corrected the chemical formulae as marked in manuscript.

Sincerely Yours,

Authors

Reviewer 4 Report

Comments and Suggestions for Authors

Comments on the Quality of English Language

Minor editing of English language required

Author Response

Dear Editor,

            We would like to thank the Reviewers for their thoughtful comments and efforts towards improving our manuscript. We appreciate their time invested in improving our work. We did our best to change manuscript according to suggestions. Please find below point-to-point responses to the Reviewers’ points.

Reviewer 4:

  1. In the introduction, please add a few more lines about the pancreatic cancer – signs and symptoms, global statistics, limitations to current therapy. Then go on to explain why organoids and GO will potentially be beneficial when translated to the clinics.

Authors response: Thank you for your valuable suggestion. We have incorporated information about the symptoms, global statistics, and limitations of current therapy for PDAC into the introduction, as indicated in the manuscript.

‘’Common signs and symptoms of PDAC, often appearing only in advanced stages, include jaundice, abdominal and back pain, unexplained weight loss, and digestive difficulties. Globally, pancreatic cancer is the seventh leading cause of cancer-related deaths, with its incidence and mortality rates closely aligning due to its aggressive nature. According to the World Health Organization, there were over 450,000 new cases and 430,000 deaths worldwide in 2020, reflecting its substantial impact on global health[2,3].

Currently, standard treatment modalities for PDAC include surgery, chemotherapy, and radiation therapy. While surgical resection offers the best chance for a cure, only a small fraction of patients is eligible for this option. The majority of pancreatic cancer cases are diagnosed at an advanced stage, rendering surgery infeasible[4]. The mainstay of treatment for advanced pancreatic cancer, chemotherapy, has limited efficacy. Drugs like gemcitabine have been the standard, but they offer only modest improvements in survival and are frequently associated with significant side effects. In addition, as it is a broadly used drug, almost all pancreatic cancer patients eventually develop resistance to this drug[5]. FOLFIRINOX (oxaliplatin, irinotecan, leucovorin, 5-fluorouracil) could directly improve overall survival(OS) rate of patients with metastatic pancreatic tumor (HR 0.76, 95% Cl 0.67–0.86, p<0.001) but had no benefit on progression-free survival(PFS)[6]. Standard radiotherapy options, which typically deliver 40 to 60 Gy in 1.8–2.0 Gy fractions, offer minimal to no survival benefit for patients with locally advanced unresectable pancreatic cancer (LAPC) who have undergone chemotherapy[7]. Its role is limited due to the proximity of the pancreas to critical organs, which increases the risk of damage to surrounding tissues. Therefore, these limitations of treatment underscore the urgent need for innovative therapeutic strategies.’’

  1. Please add a table for each of your sub-sections in section 3.3 for each of the delivery techniques you talk about. Focus on studies only from the last 3-5 years in the tables to make it easier for readers to understand how much advancement has happened in the last 3-5 years in that particular delivery method.

Authors response: Thank you for your excellent suggestion. We have added a table in "Section 3.3.1 Delivery System" to present a comprehensive list of graphene oxide (GO) delivery systems researched in recent years. This table is designed to facilitate easy understanding for readers, summarizing key advancements and applications of GO in cancer research.

‘’In this section, we discuss recent advancements in GO delivery systems for cancer research. Notably, we have compiled a list of significant recent studies on functionalized GO delivery systems (refer to Table 1). This list showcases the progress and innovations in the application of GO for targeted delivery system of cancer therapy.’’

  1. Any clinical trials that are currently active for the use of GO for pancreatic cancer? If there are more than three on going studies – please make a table of the same and add it to your ‘future’ section to strengthen it

Authors response: Thank you for your suggestion. We recognize that there are currently no clinical trials specifically involving GO for pancreatic cancer treatment, although some studies have utilized patients' blood. We have made sure to include all the existing research on GO for pancreatic cancer in Table 3 of our manuscript, ensuring a comprehensive coverage of the topic up to the present time.

  1. How about the genetic landcape of these 3D models of pancreatic cancer? Will that helpful/ useful for diagnostic/ therapy purposes? Any particular set of genes that would be of interest for sequencing? Are there other studies that have investigated this?

Authors response: Thank you very much for your interesting questions.

The genetic landscape of 3D models of pancreatic cancer is a crucial aspect of their utility in diagnostics and therapy. These models can provide valuable insights into the tumor's genetic profile, making them useful for personalized medicine approaches. For instance, genes like KRAS, CDKN2A, TP53, and SMAD4 are often mutated in pancreatic cancer and are of significant interest for genomic studies and sequencing. These genes are involved in pathways related to cell cycle regulation, apoptosis, and DNA repair, making them potential targets for therapy. Research in this domain seeks to understand how these genetic alterations contribute to tumor behavior and response to treatments.

Reference:

Driehuis, Else, et al. "Pancreatic cancer organoids recapitulate disease and allow personalized drug screening." Proceedings of the National Academy of Sciences 116.52 (2019): 26580-26590.

Tiriac, Hervé, et al. "Organoid profiling identifies common responders to chemotherapy in pancreatic cancer." Cancer discovery 8.9 (2018): 1112-1129.

Usman, Olalekan H., et al. "Genomic heterogeneity in pancreatic cancer organoids and its stability with culture." NPJ Genomic Medicine 7.1 (2022): 71.

Romero-Calvo, Isabel, et al. "Human organoids share structural and genetic features with primary pancreatic adenocarcinoma tumors." Molecular Cancer Research 17.1 (2019): 70-83.

  1. Please break down your discussion into 2-3 paragraphs to make it easier to read.

Authors response: Thank you very much for your good suggestion. We have rearranged our paragraphs.

  1. Line 820 – if organoids derived from pancreatic patients do not capture the

heterogeneity, what will? Especially considering that these patients have the diseases you are targeting? Use of cultured cells (limited by lack of better environment)? Se of iPSCs (has technical challenges)? Please think over the statement and rephrase to make what you think clear.

Authors response: Thank you for highlighting this important aspect of pancreatic cancer research. It is indeed a complex challenge to capture the full heterogeneity of the disease, especially given its intricate and diverse nature. While organoids derived from pancreatic cancer patients offer valuable insights, they may not fully encompass the entire spectrum of tumor heterogeneity. This limitation can be attributed to the selective growth of certain cell types in organoids and potential changes during the culturing process.

Exploring alternative methods can provide a broader perspective. For instance, the use of cultured cell lines, despite their limitations, can offer a controlled environment for studying specific genetic and molecular aspects of pancreatic cancer. However, these models often lack the complex tumor microenvironment and may not accurately reflect tumor behavior in vivo.

Induced pluripotent stem cells (iPSCs) present another promising avenue, capable of recapitulating a wider range of tumor characteristics due to their pluripotency. However, iPSCs come with technical challenges, such as ensuring the faithful differentiation into tumor-specific cell types and maintaining genomic stability.

We rephrased our sentences to make it clearer:

‘’Furthermore, it is indeed a complex challenge to capture the full heterogeneity of the disease, especially given its intricate and diverse nature. While organoids derived from pancreatic cancer patients offer valuable insights, they may not fully encompass the entire spectrum of tumor heterogeneity. This limitation can be attributed to the selective growth of certain cell types in organoids and potential changes during the culturing process.[298-300].’’

  1. Overall, please limit a paragraph to 7-10 lines to make it easier to read. Otherwise the manuscript appears extremely text heavy and difficult for readers.

Authors response: Thank you very much for your useful suggestion. We rearranged our paragraphs as indicated in the revised manuscript.

Sincerely Yours,

Authors